# Cell-lineage controlled epigenetic regulation in glioblastoma stem cells determines functionally distinct subgroups and predicts patient survival

Xi Lu [1,8], Naga Prathyusha Maturi [2,8], Malin Jarvius[3,4], Irem Yildirim [2], Yonglong Dang[1,5], Linxuan Zhao[1], Yuan Xie[2,6], E-Jean Tan [2,7], Pengwei Xing[1], Rolf Larsson[3], Mårten Fryknäs[3], Lene Uhrbom [2,9✉] & Xingqi Chen [1,9✉]

There is ample support for developmental regulation of glioblastoma stem cells. To examine how cell lineage controls glioblastoma stem cell function, we present a cross-species epigenome analysis of mouse and human glioblastoma stem cells. We analyze and compare the chromatin-accessibility landscape of nine mouse glioblastoma stem cell cultures of three defined origins and 60 patient-derived glioblastoma stem cell cultures by assay for transposase-accessible chromatin using sequencing. This separates the mouse cultures according to cell of origin and identifies three human glioblastoma stem cell clusters that show overlapping characteristics with each of the mouse groups, and a distribution along an axis of proneural to mesenchymal phenotypes. The epigenetic-based human glioblastoma stem cell clusters display distinct functional properties and can separate patient survival. Cross-species analyses reveals conserved epigenetic regulation of mouse and human glioblastoma stem cells. We conclude that epigenetic control of glioblastoma stem cells primarily is dictated by developmental origin which impacts clinically relevant glioblastoma stem cell properties and patient survival.

[1] Department of Immunology, Genetics and Pathology, Uppsala University, SE-75108 Uppsala, Sweden. [2] Department of Immunology, Genetics and Pathology, Uppsala University and Science for Life Laboratory, Rudbeck Laboratory, SE-75185 Uppsala, Sweden. [3] Department of Medical Sciences, Cancer Pharmacology and Computational Medicine, Uppsala University and Science for Life Laboratory, SE-75185 Uppsala, Sweden. [4] Present address: Department of Pharmaceutical Biosciences and Science for Life Laboratory, Uppsala University, Box 591, SE-751 24 Uppsala, Sweden. [5] Present address: Laboratory of Molecular Neurobiology, Department of Medical Biochemistry and Biophysics, Karolinska Institutet, SE-171 77 Stockholm, Sweden. [6] Present address: Shaanxi Normal University, College of Life Sciences, Xi'an 710119, China. [7] Present address: Department of Organismal Biology, Uppsala University, Norbyvägen 18A, SE-75236 Uppsala, Sweden. [8] These authors contributed equally: Xi Lu, Naga Prathyusha Maturi. [9] These authors jointly supervised this work: Lene Uhrbom, Xingqi Chen. ✉email: lene.uhrbom@igp.uu.se; xingqi.chen@igp.uu.se

Glioblastoma (GBM) is one of the most aggressive cancers and the most frequent and lethal primary malignant brain tumor[1]. Standard therapy of care includes maximal-safe surgical resection, concomitant chemo- and radiotherapy, and adjuvant chemotherapy, yet the 2-year survival is 18.5%[1]. Treatment resistance is explained by extensive genetic and epigenetic tumor cell heterogeneity of GBM, both with regard to intertumor heterogeneity[2–4] and intratumor heterogeneity at different regions[5] and in individual cells[6–9]. Large efforts have been done to converge GBM heterogeneity into biologically and clinically relevant subgroups of GBM. Transcriptome-based stratifications have produced three major isocitrate dehydrogenases 1 and 2 (*IDH1* and *IDH2*) wild-type (wt) GBM subtypes, also called The Cancer Genome Atlas (TCGA) subtypes: proneural (PN), classical (CL), and mesenchymal (MS)[2,10,11]. Studies of patient-derived GSC cultures, clonal derivatives, and single cells have shown the presence of a PN to MS differentiation axis with plasticity of the states[8,12,13], and a comprehensive GBM single-cell analysis has uncovered additional and dynamic cellular states in GBM tumors[7]. The GBM epigenome has been most frequently analyzed by DNA methylation profiling[3,4,14] and methylomes have proven prognostically more useful than transcriptomes to predict patient survival[3,4], demonstrating the importance of understanding epigenetic regulation in GBM. The active chromatin landscape of GBM has been investigated with chromatin immunoprecipitation sequencing (ChIP-seq) of acetylated lysine 27 on histone H3 (H3K27ac) in a collection of primary tumors and GSC cultures[15], and by the assay for transposase-accessible chromatin using sequencing (ATAC-seq)[8,16,17], which have uncovered subgroups of GBM suggested to be regulated by different sets of transcription factors (TFs).

Several studies have implied a connection between GBM molecular subgroups and developmental origin[2,18]. Methylation profiling has proven particularly useful to connect primary tumors with their tissue of origin[19] and has been used to separate GBM with higher resolution than gene expression[4,20]. We have shown by experimental modeling of GBM that developmental state and age of the cell of origin could affect its vulnerability to GBM development[21], and that it shaped the phenotype of the resulting GBM stem cells (GSCs)[21,22]. Tumors were induced by the same oncogenic events in three different mouse cell lineages which produced contrasting tumor cell phenotypes with regard to malignancy and drug sensitivity, where a more differentiated origin promoted a less tumorigenic but more drug resistant mouse GSC (mGSC) phenotype[22]. Through a cross-species GSC-based stratification approach applying the mouse cell of origin (MCO) gene signature of differentially expressed genes on a large collection of human GSC (hGSC) cultures, we found that developmental origin could be used to stratify functionally distinct groups of patient-derived GSC cultures[22]. A recent similar cross-species approach has further corroborated the importance of cell lineage origin in GBM[23]. In all this has demonstrated that intertumor heterogeneity to a large extent is shaped by the intrinsic properties of the GBM cell of origin which result in highly dynamic GSCs that basically evade all current therapies.

Here we show a cross-species epigenome analysis where the chromatin accessibility landscape of 9 mGSC cultures of defined developmental origin and 60 IDH wt hGSC cultures are analyzed with high-sensitivity ATAC-seq[24]. We relate the results to a range of molecular and functional data and show that genome-wide chromatin accessibility separates both mouse and human GSC cultures into three functionally distinct subgroups. The mGSC groups are divided by developmental origin and display shared molecular features with each of the three hGSC groups along a PN to MS axis. Cross-species analyses support the hGSC subgroups to be cell lineage controlled and show a conservation of enriched TF motifs in the differential chromatin-accessible regions. Importantly, the ATAC-seq-based stratification can separate patients with significantly different survival pointing to the ability of this analysis to distinguish clinically relevant tumor cell properties.

## Results

**Chromatin accessibility in mouse GSCs predicts developmental origin.** We performed ATAC-seq analysis of nine previously established mGSC cultures (mGC1$_{GFAP}$: SC81, SC83, SC84; mGC2$_{NES}$: SC50, SC52, SC64; mGC3$_{CNP}$: SC37, SC74, SC112) derived from GBMs induced by the same oncogene (PDGFB) in three different cell lineages in adult tv-a transgenic mice (*Gfap/tv-a;Arf−/−*, *Nes/tv-a;Arf−/−*, and *Cnp/tv-a;Arf−/−*, respectively)[22] (Fig. 1a and Supplementary Data 1). The cell of origin in these mice had previously been deduced to be a neural stem cell (NSC)-like cell in *G/tv-a* mice, an astrocyte precursor cell-like cell in *N/tv-a* mice and an oligodendrocyte precursor cell (OPC)-like cell in *C/tv-a* mice[22], and cell cultures of different origin had displayed distinct functional, phenotypic, and transcriptomic properties[22]. The ATAC-seq data from all samples were determined to be of high quality based on analyses of enrichment of sequence reads at transcription start sites (TSS) (Fig. 1b), fraction of reads in peaks (FRiP) (Fig. 1c) and reproducibility among the technical replicates (Supplementary Fig. 1a). Genomic annotation of ATAC-seq data displayed some variation across cultures, but showed high reproducibility between replicates (Supplementary Fig. 1b). In all this supported a high quality of the mouse ATAC-seq data.

Previous principal component analysis (PCA) of gene expression array data from the same mGSC cultures had shown a clear separation based on developmental origin[22]. PCA analysis of global ATAC-seq data also distinguished cell of origin groups and produced a clear separation between NSC and GSC cultures (Fig. 1d). To understand the underlying molecular regulation of the mGSC groups we extracted the differentially enriched ATAC-seq peaks (Log 2 (Fold change (FC)) > 1, false discovery rate (FDR) < 0.05) of each group (Fig. 1e and Supplementary Data 2), which produced 819 peaks for mGC1$_{GFAP}$, 1161 peaks for mGC2$_{NES}$ and 95 peaks for mGC3$_{CNP}$. The differential ATAC peaks were annotated to their nearest genes (Supplementary Data 2), where some examples are indicated for each of the mGSC groups (Fig. 1e). Representative genome tracks for each group validated the differential chromatin accessibility of annotated genes (Fig. 1f). Many genes of the mGC2$_{NES}$ group showed a clear mesenchymal character (e.g., *Cd44*, *Bmp7*, *Tgfb2*) which is in line with the previous finding that mGC2$_{NES}$ cells were most closely related to the MS GBM subtype, while both the mGC1$_{GFAP}$ and mGC3$_{CNP}$ cells were most similar to PN GBM[22]. To widen our understanding of the characteristics of each mGSC group we performed Gene ontology (GO) enrichment analysis of the annotated genes (Supplementary Fig. 1c–f). The top-15 GO terms selected by strength and ranked by FDR were for mGC1$_{GFAP}$ mainly related to voltage gated ion channels and RAS GTPase activity for molecular function (Supplementary Fig. 1c), and to synaptic functions and adult responses to neuronal stimuli for biological processes (Supplementary Fig. 1d). For mGC2$_{NES}$ the majority of molecular functions were involved in protein binding and RNA polymerase II DNA binding (Supplementary Fig. 1e) and for biological processes the GO terms were centered around epithelial and mesenchymal cells (Supplementary Fig. 1f). For mGC3$_{CNP}$ the number of genes were small and there were no significant GO results. Taken together, also these results supported a mesenchymal character of the mGC2 cells and a neural character of mGC1 cells.

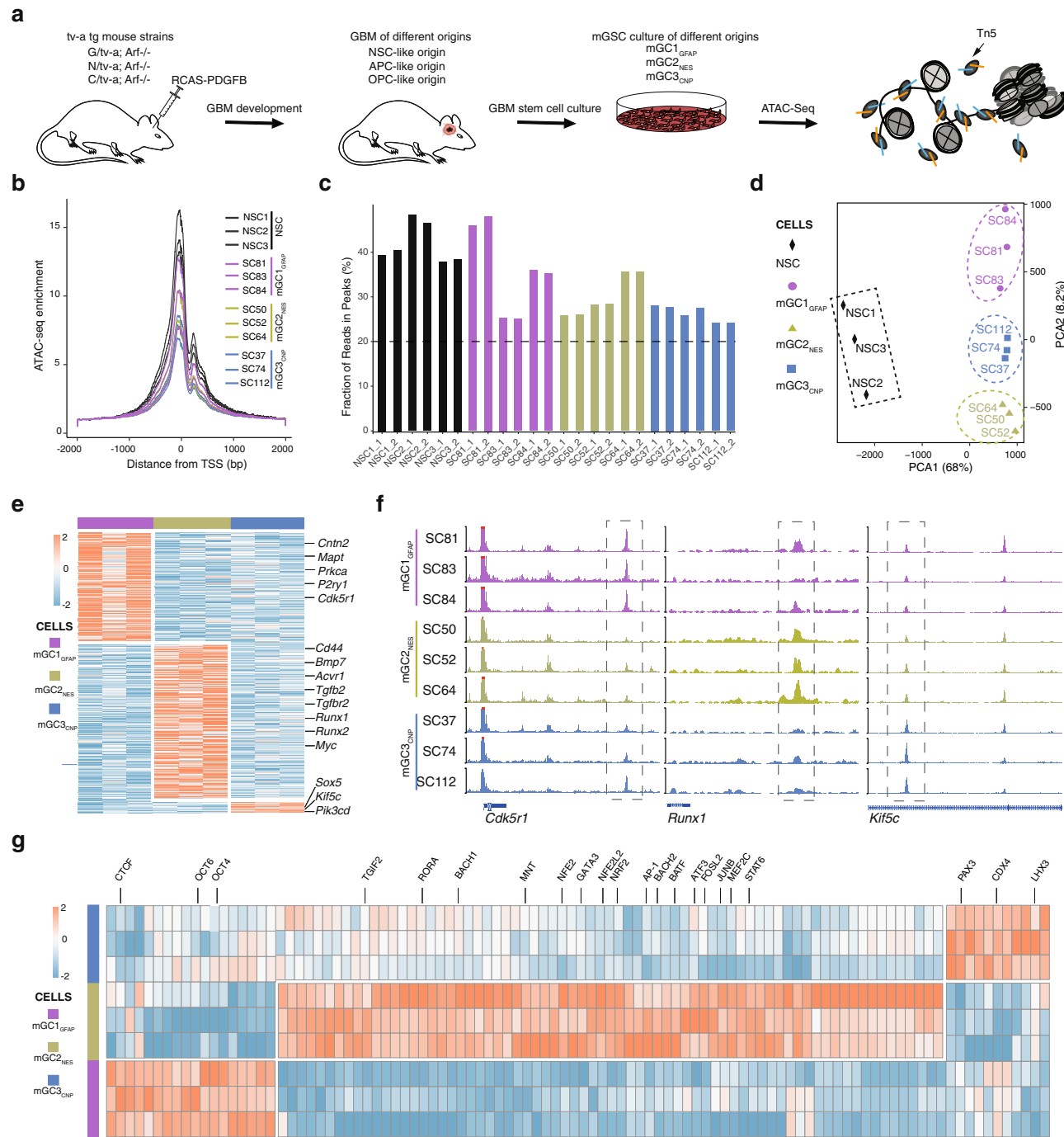

**Fig. 1 The chromatin accessibility landscape of mouse GSC cultures can predict cell of origin. a** Overview of mouse GBM models and GSC cultures of different origin that were analyzed by ATAC-seq. Schematic figures were produced by the authors. **b** Enrichment of ATAC-seq reads at transcription start sites (TSS) from the mGSC cultures. **c** Fraction of reads in peaks (FRiP) in duplicate samples of mouse NSC and GSC cultures. Dashed line shows minimum FRiP score required for further analyses of that sample. **d** Principal component analysis of ATAC-seq data from mouse NSC and GSCs samples. Dashed ovals show same origin mGSC samples. Dashed square show control NSC samples. **e** Heatmap of unique accessible chromatin regions of each mGSC cell of the origin group. Examples of genes annotated to the chromatin regions are indicated. **f** Genome browser tracks of ATAC-seq data for each mGSC sample of *Cdk5r1*, *Runx1*, and *Kif5c*. **g** Heatmap of mGSC group-unique, significantly enriched TF motifs. Examples of TFs are indicated. Source data are provided as a Source Data file for Fig. 1b–e, g.

To discover the differential regulatory motifs between cultures of different origin we performed TF motif analysis of the unique accessible chromatin regions and extracted the unique TFs among the mGSC groups (Fig. 1g and Supplementary Data 3). In line with having the highest number of unique ATAC regions mGC2_NES samples also produced the largest number of uniquely

enriched TF motifs. Overall, the result suggested different regulatory circuits between the mGSC groups, which was further sustained when extracting all significantly enriched and variable TF motifs of each group (Supplementary Fig. 1g–i). These were highly overlapping with the unique TF motifs (Fig. 1g) but in this analysis all significant TFs for each group were included which

produced a clear overlap between mGC1$_{GFAP}$ and mGC3$_{CNP}$, and mGC3$_{CNP}$ and mGC2$_{NES}$, while clearly separating mGC1$_{GFAP}$ and mGC2$_{NES}$.

In summary, the ATAC-seq analysis of mGSC cultures produced, in accordance with our previous gene expression analysis, distinct cell of origin groups. Analysis of TF motifs in differential accessible chromatin supported distinct differences between mGC1$_{GFAP}$ and mGC2$_{NES}$ while there was some overlap between mGC3$_{CNP}$ and mGC1$_{GFAP}$ or mGC2$_{NES}$. This proposed that although all mGSC cultures have the same driver mutations (PDGFB overexpression and p19$^{Arf}$ deletion) there are important developmentally inherited mechanisms that regulate their previously shown[22] different phenotypes.

**Heterogeneous chromatin accessibility across 60 patient-derived GSC cultures.** Next, we investigated the chromatin accessibility landscape in GSC cultures of our local human GBM cell culture (HGCC) biobank[22,25,26]. We performed ATAC-seq on 60 patient-derived IDH wt GSC cultures (Fig. 2a and Supplementary Data 1). We applied established and stringent criteria for the ATAC-seq data processing and could show high Pearson correlation coefficient (0.8–0.98) for technical replicates, TSS enrichment scores above 3.8, and FRiP of all duplicate samples at or above 20%[27] (Supplementary Fig. 2a–c). We performed a saturation analysis using random sampling non-linear regression to analyze the fraction of all predicted accessible chromatin regions that we could expect to detect with 60 cultures (Fig. 2b). This showed that our sample size was large enough and that all predicted regions of accessible chromatin would be detected with 47 cultures. In total, we had captured 323526 ATAC peaks from our hGSC cohort. To obtain an overview of the global chromatin accessibility landscape of the samples we identified all unique chromatin-accessible regions in the entire data and calculated for each region the number of samples it was present in (Fig. 2c). A large proportion (25.4%) of ATAC peaks was only detected in one hGSC culture, and just 1.5% of the accessible regions were common to all 60 cultures. Nonetheless, unique chromatin-regions showed increased openness the more frequently they were present across the samples (Supplementary Fig. 2d). SOX2 showed an overall high chromatin accessibility across the cohort (Fig. 2d) in line with previous data showing SOX2 expression in all HGCC cultures investigated[28]. Yet, individual cultures showed a clear variability in chromatin openness of this locus (Fig. 2d). Inter-culture heterogeneity was further sustained by separate analyses of promoter ($-1$ kbp to $+100$ bp of TSS) and non-promoter (also called distal regulatory element (DRE)) regions of GSC metamodule genes[7] and MCO genes[22] (Fig. 2e and Supplementary Fig. 2e). Also structural genomic annotation of the ATAC data showed a clear variation in chromatin-openness among the 60 cultures (Fig. 2f). Taken together, this demonstrated that our cohort of 60 patient-derived GSC cultures displayed a highly heterogeneous chromatin accessibility landscape in line with the high GBM inter-patient diversity.

**Chromatin-accessibility robustly identifies three clusters of patient-derived GSC cultures.** To identify unifying features of the hGSC cohort we performed non-negative matrix factorization (NMF) analysis on the ATAC-seq data, which produced three clusters: ATAC60-C1 ($n = 22$), ATAC60-C2 ($n = 16$), and ATAC60-C3 ($n = 22$) (Supplementary Fig. 3a, b and Supplementary Data 1). A large part of the HGCC cultures had previously been classified based on gene expression according to the TCGA subtypes[28] and with the MCO gene signature (MCO1–3)[22]. To compare the ATAC-seq clusters with the TCGA and MCO classifications we excluded hGSC samples lacking such

information and re-analyzed 50 samples with NMF. This produced, again, three clusters: ATAC50 C1 ($n = 19$), ATAC50 C2 ($n = 14$), and ATAC50 C3 ($n = 17$) (Fig. 3a and Supplementary Fig. 3c). Comparing ATAC50 to ATAC60 clusters showed that only three samples had changed cluster in ATAC50 (Supplementary Fig. 3d and Supplementary Data 1), which showed a robustness of the chromatin accessibility-based clustering. From hereon we focus mainly on the ATAC50 classification.

When we compared the ATAC50 clusters with the TCGA subtypes there was little overlap (Fig. 3b). The majority of PN and CL cultures were in C1 while MS cultures basically were divided between C2 and C3. Comparing to MCO showed a higher degree of overlap (Fig. 3c), likely reflecting the relation between developmental origin and epigenetic state of GSC.

Next, we extracted the unique chromatin-accessible regions for each ATAC50 cluster with DESeq2 (Log 2 (FC) > 1, FDR < 0.01, peak average intensity >30, and coefficient of variance <0.2; Fig. 3d). By this we identified 4023 regions in C1, 5547 in C2, and 949 in C3 (Fig. 3d and Supplementary Data 4). The genomic features of all chromatin-accessible regions from each ATAC50 cluster were annotated using the chromatin state discovery and characterization software (ChromHMM)[29] (Fig. 3e and Supplementary Fig. 3e). There was a diverse distribution of chromatin states with some marked differences between ATAC50 clusters. C1 had the largest proportion of active promoter regions (H3K4me3 and H3K27ac), C2 occupied a higher proportion of active regions (H3K27ac), and C3 had a higher frequency of weak enhancer regions (H3K4me1). Common to all three clusters was that the combined proportion of non-promoter regions (strong and weak enhancer regions) constituted the biggest proportion of chromatin states, clearly larger compared to the distribution in the whole ATAC-seq data (Fig. 3e). This indicated, as for mGSCs, that DRE regions were central in defining the ATAC50 clusters. To test our hypothesis, we performed Pearson correlation hierarchal clustering of all ATAC peaks, of DRE regions only, and of promoter regions only (Supplementary Fig. 3f). All ATAC peaks and DRE peaks displayed a similar dynamic range of chromatin openness and cluster patterns, while the dynamic range of promoter regions was smaller which supported our assumption. NMF clustering of DRE ATAC peaks (Fig. 3f and Supplementary Fig. 3g) produced almost identical clusters as ATAC50 (Fig. 3g). Clustering promoter regions resulted in three clusters (Supplementary Fig. 3i) that were entirely different and non-overlapping with ATAC50 (Supplementary Fig. 3j). Collectively, our analyses showed that chromatin accessibility could robustly stratify hGSC cultures and clusters were predominantly dictated by the DRE regions. The high correspondence of the ATAC50 clusters with the MCO stratification implied an important role of cell lineage-controlled gene regulation of human GSC cultures.

**ATAC50 clusters are phenotypically distinct.** To phenotypically characterize the ATAC50 clusters we first used hGSC gene expression array data[22] and analyzed the 256 GSC meta module genes[7] across the 50 cultures (Fig. 4a and Supplementary Fig. 4). While only 53 genes showed a significant difference between the clusters (Fig. 4b) there were clear differences comparing global meta module gene expression (Fig. 4a). C1 showed significantly higher expression of NPC1, NPC2, and OPC genes compared to both C2 and C3, significantly higher expression of AC genes compared to C2, and significantly lower expression of MES genes compared to both C2 and C3. Thus, C1 and C2 were always at the end of the spectrum with C3 in the middle. Notably, C3 showed significantly higher expression of NPC1, OPC, and AC genes and significantly lower expression of MES2 genes compared to C2.

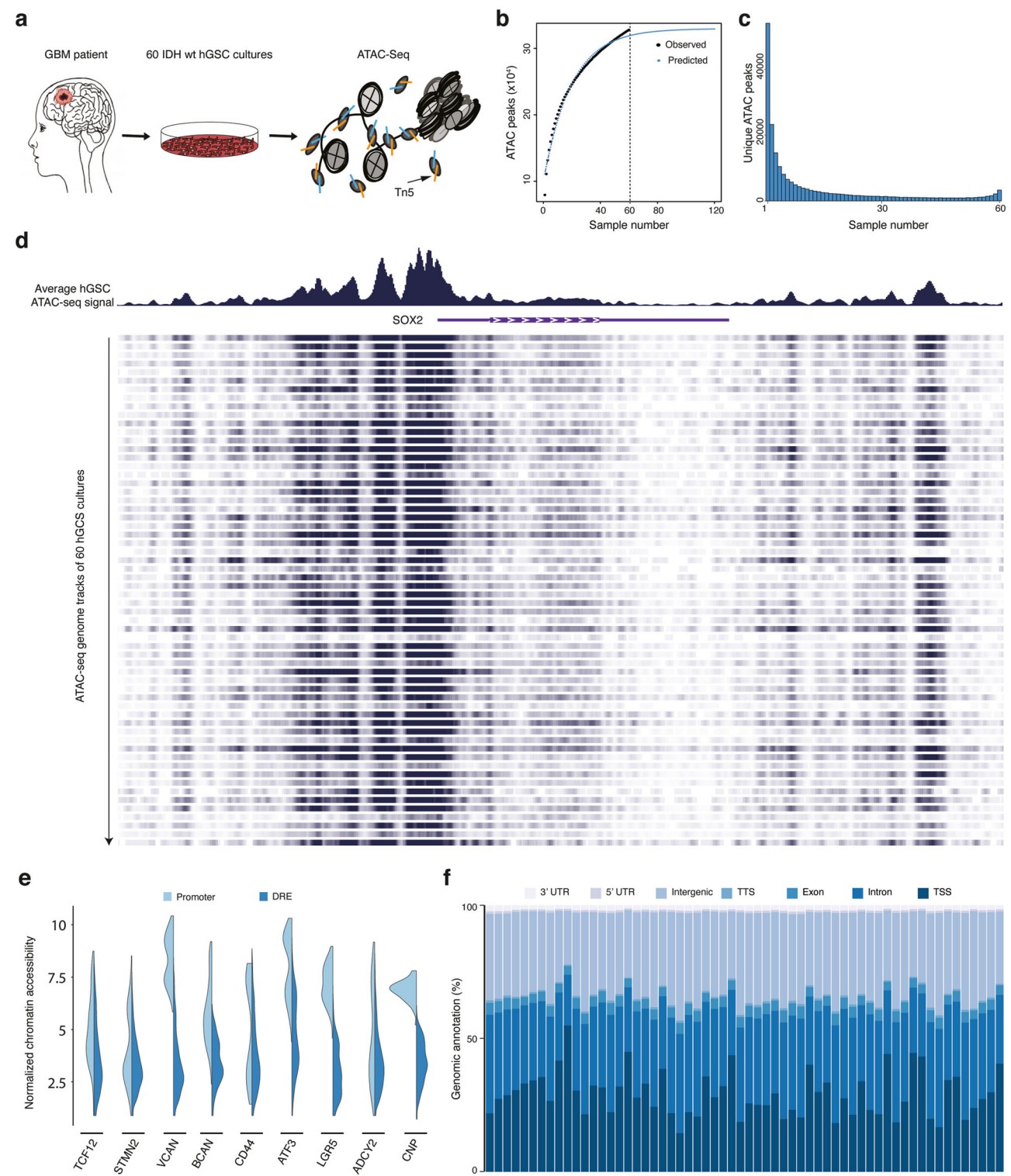

**Fig. 2 Heterogeneous chromatin accessibility across 60 patient-derived GSC cultures. a** Overview of 60 patient-derived IDH wild-type GSC cultures analyzed by ATAC-seq. Schematic figures were produced by the authors. **b** Saturation analysis using a non-linear model. Number of predicted accessible chromatin regions in GBM (blue dotted line), and number of observed accessible chromatin regions in our 60 hGSC samples (black dotted line). **c** Histogram of the distribution of unique ATAC peaks in the hGSC cohort. **d** Genome browser tracks of ATAC-seq signals at *SOX2*. Top panel shows the average genome track of all 60 samples. Bottom panel shows individual results. **e** Violin plots of cohort-wide distribution of chromatin accessibility at promoters and DRE regions of some GBM meta module and cell lineage-relevant genes. **f** Genomic annotation of ATAC peaks in each hGSC sample. Source data are provided as a Source Data file for Fig. 2b, c, e, f.

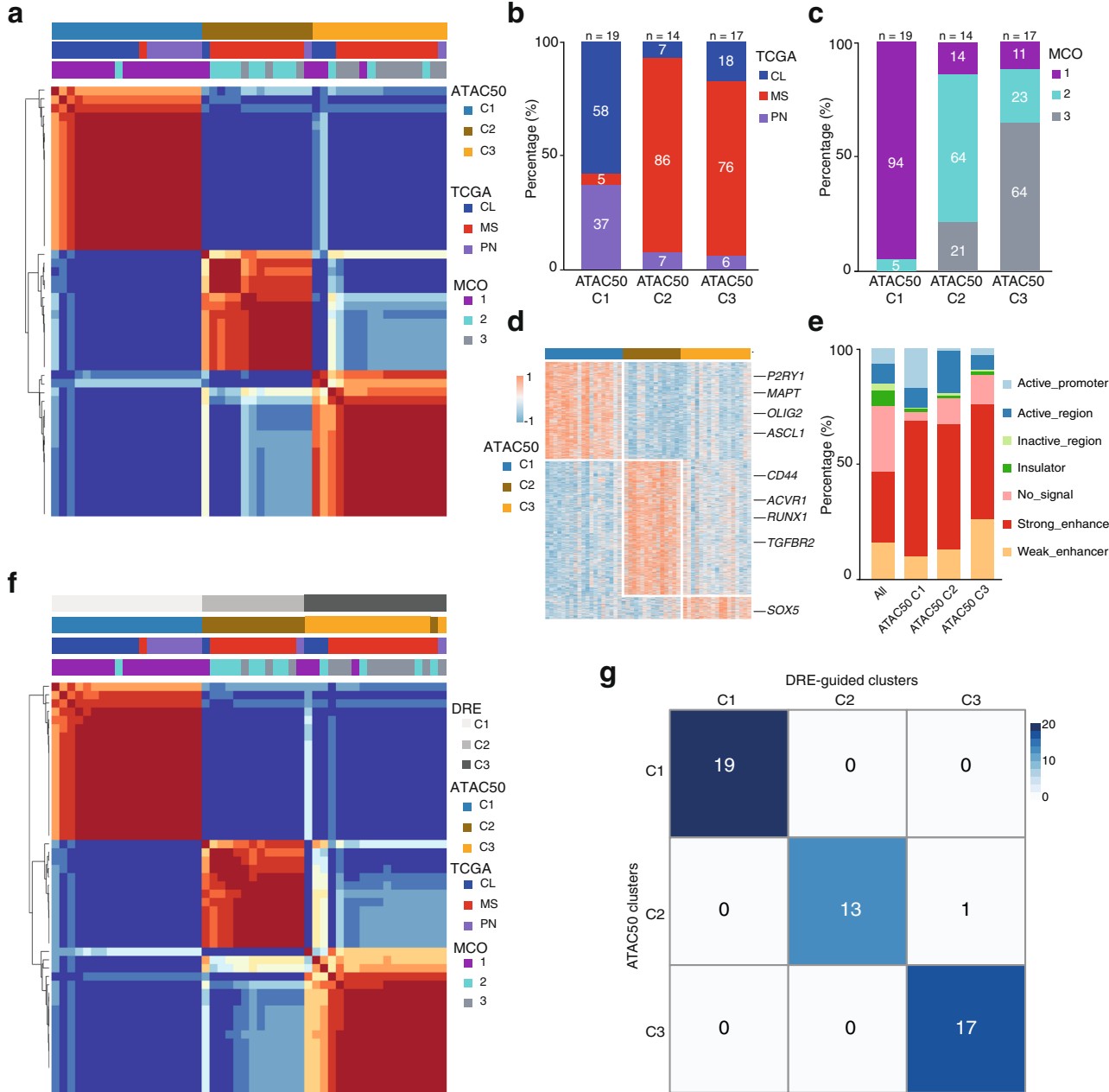

**Fig. 3 Chromatin-accessibility robustly identifies three clusters of patient-derived GSC cultures. a** NMF cluster analysis of ATAC-seq data from 50 patient-derived GSC cultures. **b** Distribution of TCGA subtypes among ATAC50 clusters. **c** Distribution of MCO subgroups among ATAC50 clusters. **d** Heatmap of unique ATAC peaks of each ATAC50 cluster. Examples of genes annotated to the chromatin regions are indicated. **e** Genomic annotation of all ATAC peaks in the hGSC cohort and unique ATAC peaks of each ATAC50 cluster. **f** NMF cluster analysis of DRE ATAC peaks from 50 human GSC cultures. **g** Overlap of ATAC50 clusters with DRE-guided clusters. Source data are provided as a Source Data file for Fig. 3a–g.

This suggested that ATAC50 clusters were separated along a gradient of GSC states with C1 being progenitor cell-like, C2 being mesenchymal-like, and C3 being intermediate.

Next, we analyzed chromatin openness of metamodule genes in promoter regions (Fig. 4c) and DRE regions (Fig. 4d). The openness of DRE regions was significantly different between all clusters in all metamodules whereas promoter regions showed less distinct differences for NPC1, NPC2, OPC, and AC metamodules and were nonsignificant for MES1 and MES2. This is in line with the dominant role of DRE regions to separate the ATAC50 clusters (Fig. 3f). Although the differences between ATAC50 groups in most of the comparisons in Fig. 4a, c, d were

statistically significant the box plots were still highly overlapping reflecting the extensive intertumor heterogeneity of GBM.

Of the metamodule genes with significant different gene expression (Fig. 4b) the majority (39) were higher expressed in C1 and belonged to the NPC1, NPC2, OPC, and AC modules. C2 showed a higher expression of some MES1 and MES2 genes, and C3 cultures displayed higher expression of a few NPC1, NPC2, and AC genes. To investigate the chromatin openness of DRE regions of these genes we first linked all human ATAC DRE regions with their nearest gene through a peak-to-gene linking prediction analysis[27] (Supplementary Fig. 5a and Supplementary Data 5). This was used to compare ATAC peaks of the significant

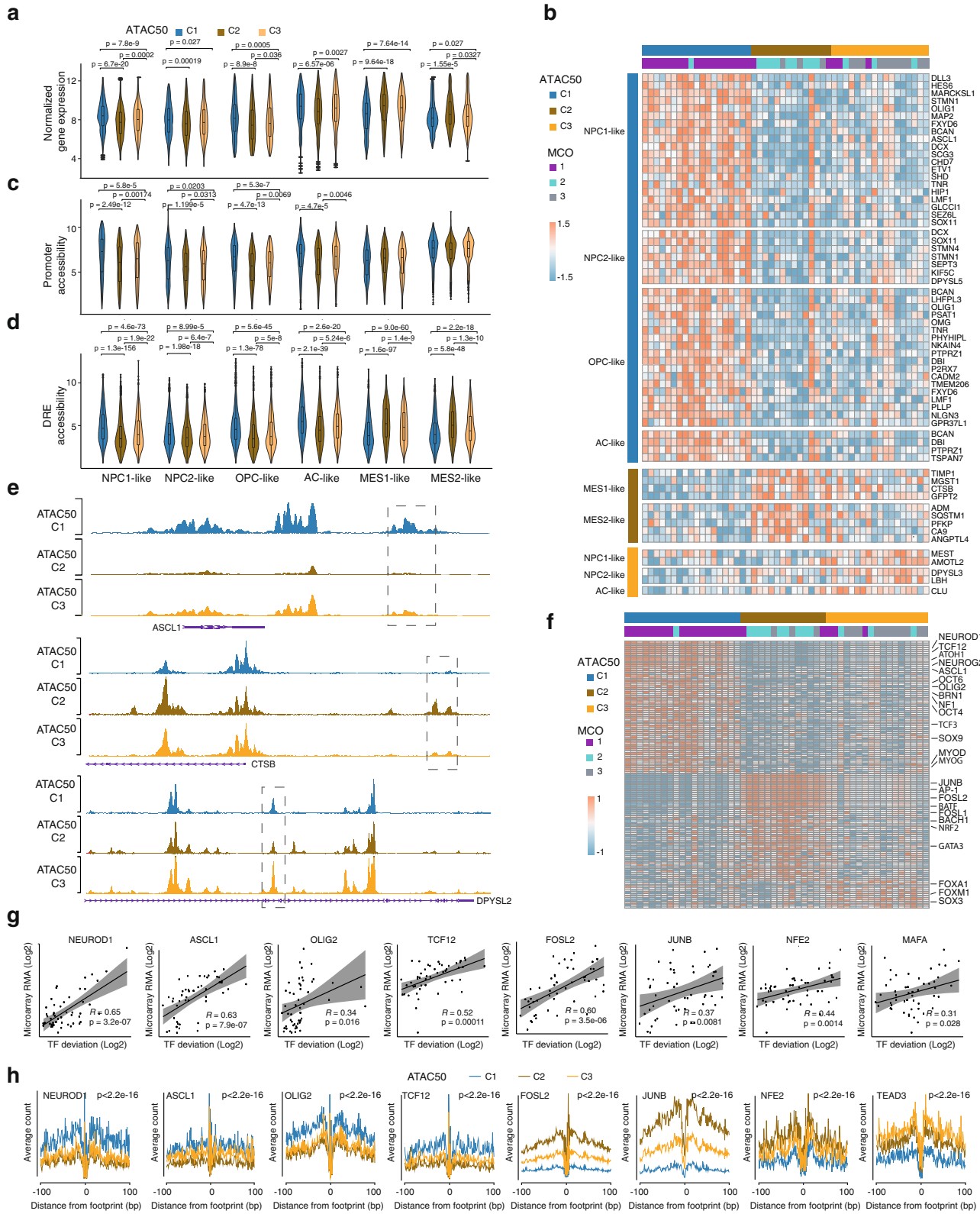

genes (Fig. 4b) between clusters (Supplementary Data 5) which showed that 47% (25 of 53) had significantly different chromatin accessibility in DRE regions (Fig. 4e and Supplementary Data 5), compared to only 7 of 53 for promoter regions, supporting the importance of DREs in regulating ATAC50 clusters.

To investigate underlying mechanisms controlling the ATAC50 clusters, we performed TF motif enrichment analysis

on significantly different chromatin-accessible regions. Of the top-50 most variable TFs motifs the majority were bZIP ($n = 14$) or bHLH ($n = 15$) motifs (Supplementary Fig. 5b and Supplementary Data 6). There was a clear and inverse enrichment when comparing ATAC50 C1 and C2, with bHLH motifs being most common in C1 and bZIP motifs being most common in C2. GSCs maintained in stem cell media have been shown to enrich for

**Fig. 4 ATAC50 clusters are phenotypically distinct. a** Comparison between ATAC50 clusters of average gene expression of all genes in each GSC meta module. **b** Meta module genes with significantly different gene expression between ATAC50 clusters. **c** Comparison between ATAC50 clusters of average chromatin accessibility at promoters of all genes in each GSC meta module. **d** Comparison between ATAC50 clusters of average chromatin accessibility at DRE regions of all genes in each GSC meta module. The box plots in the violin plots (**a**, **c**, **d**) show the minima (bottom dot), the maxima (top dot), the median (middle line), and the first and third quartiles (boxes), whereas the whiskers show 1.5× the interquartile range IQR above and below the box. For GSC cell line numbers: ATAC50 C1 ($n = 19$), ATAC50 C2 ($n = 14$), and ATAC50 C3 ($n = 17$). In plots (**a**, **c**, **d**), 256 GSC metamodule genes were used to compared. Two-sided Welch's $t$-test was performed on all pair-wise comparisons in (**a**, **c**, **d**), significant differences are indicated with $p$ value. **e** Genome browser tracks of the average ATAC-seq signal in each ATAC50 cluster for *ASCL1, CTSB, DPYSL2*. **f** Heatmap of significantly enriched cluster-unique TF motifs. **g** Scatter plots of TF motif chromatin accessibility ($x$-axis, normalized chromatin openness) and TF gene expression ($y$-axis, microarray data, counts per million reads mapped (CPM)). All significantly enriched TF motifs in (**f**) were analyzed and those with a significant positive correlation are shown. $R =$ Pearson correlation, $p$ calculated by two-sided Welch's $t$-test. Ninety-five percent confidence intervals are indicated with shaded areas. **h** TF footprint analysis of cluster-specific ATAC50 peaks. Friedman–Nemenyi test was performed, $p < 0.05$ for all. The $p$ value for each TF is labeled. Source data are provided as a Source Data file for Fig. 4a–d, f–h.

bHLH TFs while serum media enriched for bZIP TFs[30] corroborating different state identities of C1 and C2 cultures. To find distinctive features of each cluster we extracted the significantly enriched cluster-specific TF motifs (Fig. 4f and Supplementary Data 7). This identified 64 uniquely enriched motifs in C1, 51 in C2, and 13 in C3. Among the TF motifs in C1 many were regulators of neural development with strong connections to GBM such as TCF12 (refs. [31–34]), ASCL1 (ref. [35]), OLIG2 (ref. [36]), and SOX9 (ref. [37]). Dominant TF motifs enriched in C2 were AP-1 complex motifs of the JUN, FOS, ATF, and MAF families and motifs of the MAF dimerizing proteins NRF2, BACH1, and BACH2, which have also been associated with cancer progression and metastasis[38] and are plausible candidates to regulate the mesenchymal features of C2 cultures. Since C3 was intermediate to C1 and C2, there were fewer uniquely enriched TF motifs in this cluster. Among them were GBM-associated FOXM1, FOXA1, and SOX3 (Fig. 4f)[31–34,39]. When analyzing the relationship of TF motif openness and TF gene expression we found a positive correlation for several TFs (Fig. 4g) which strengthened their involvement in shaping the cluster phenotypes. To further analyze TF regulation we performed TF footprint analysis of differential ATAC peaks, which is a computational method to predict TF binding (Fig. 4h and Supplementary Fig. 5c). We found that ATAC50 C1 showed significantly higher occupancy of OLIG2, TCF12, ASCL1, and NEUROD1 compared to C2 and C3, while C2 and C3 showed significantly higher occupancy for NF-E2, JUNB, and FOSL2 compared to C1. Among the C3-uniquely enriched TF motifs there were no significant footprints, but among the top-50 variable TF motifs TEAD3 showed significantly higher occupancy for C3 compared to C1 (Fig. 4h). In all, the TF motif enrichment, TF gene expression, and TF occupancy analyses confirmed the phenotypic differences and revealed distinct epigenetic regulation of the ATAC50 clusters.

**ATAC50 classification produce functional separation of hGSC cultures.** Since the transcriptome and ATAC-seq defined mGSC groups had displayed different functional properties[22], we asked if the ATAC50 clusters also would. We investigated essential GSC properties in 16 C1, 8 C2, and 13 C3 cultures (Fig. 5), of which the majority had overlapping MCO and ATAC50 classifications (Supplementary Data 1). As a reference we also analyzed all data by grouping the cultures according to TCGA subtype (Supplementary Fig 6 and Fig. 6b). We first performed consecutive sphere-forming assays under clonal conditions (Fig. 5a and Supplementary Fig. 6a). C1 cultures displayed the highest sphere-forming ability while there was no significant difference between C2 and C3 cultures, although C3 cultures produced a higher average number of spheres (Fig. 5a). TCGA grouping also

produced significant differences between the MS subtype and CL or PN (Supplementary Fig. 6a). Extreme limiting dilution assay (ELDA) is considered a more reliable and objective measurement of self-renewal and showed that C1 cultures had a significantly higher self-renewal capacity compared to both C2 and C3. However, this method also captured a significant difference between C2 and C3 (Fig. 5b). With this method TCGA groups only captured a difference between MS and the other two subtypes (Supplementary Fig. 6b). Cell proliferation was measured by BrdU incorporation and here C1 cultures showed significantly higher proliferation compared to both C2 and C3 cultures (Fig. 5c). For this trait TCGA classification could distinguish a significantly lower proliferative capacity of MS cells compared CL and PN (Supplementary Fig. 6c). Tumor cell invasion was analyzed with the spheroid collagen gel invasion assay. Here we found that C2 cultures were significantly more invasive than both C1 and C3 cultures (Fig. 5d), while TCGA subtypes could separate the MS and CL groups only (Supplementary Fig. 6d). In all, the functional characteristics were in accordance with the stem and progenitor cell-like molecular phenotype of C1 cultures, the mesenchymal-like phenotype of C2 culture and the mixed molecular phenotype of the C3 cultures. It also showed that ATAC50 was superior to TCGA in predicting two key features of GSCs, self-renewal and invasion.

We also analyzed the drug response phenotype of 11 C1, 7 C2, and 10 C3 cultures by measuring cell viability after 72 h exposure to a collection of 28 anticancer drugs at seven different concentrations (Fig. 5e–g and Supplementary Data 8). This produced dose-response curves that were converted to area under the curve (AUC) measures that were compared pair-wise between clusters. There was a clear overall higher sensitivity of C1 cultures to the compounds compared to both C2 (Fig. 5e) and C3 (Fig. 5f) cultures. All drugs that produced a significantly different response between C1 and C2 or C3 cultures were more effective in C1 cultures. These comparisons identified two compounds as particularly efficient for C1 cultures, Melflufen (alkylating), and PD173074 (FGFR1 inhibitor). When comparing C2 to C3 cultures, C3 cultures were clearly, overall, more sensitive to the tested drugs (Fig. 5g). However, C3 cultures showed a significantly higher resistance to two drugs, 5-azacytidine and 6-thioguanine, compared to both C1 and C2 cultures (Fig. 5f, g). When we compared drug responses between TCGA groups the result was less pronounced (Supplementary Fig. 6e–g). We found that PN cultures had an overall higher sensitivity to these compounds compared to MS cultures (Supplementary Fig. 6e) while there were small differences between PN and CL (Supplementary Fig. 6f), and MS and CL (Supplementary Fig. 6g). The distinct drug response phenotypes of the ATAC50 clusters suggested that cell lineage dependencies are important to account for when developing therapeutic strategies for GBM.

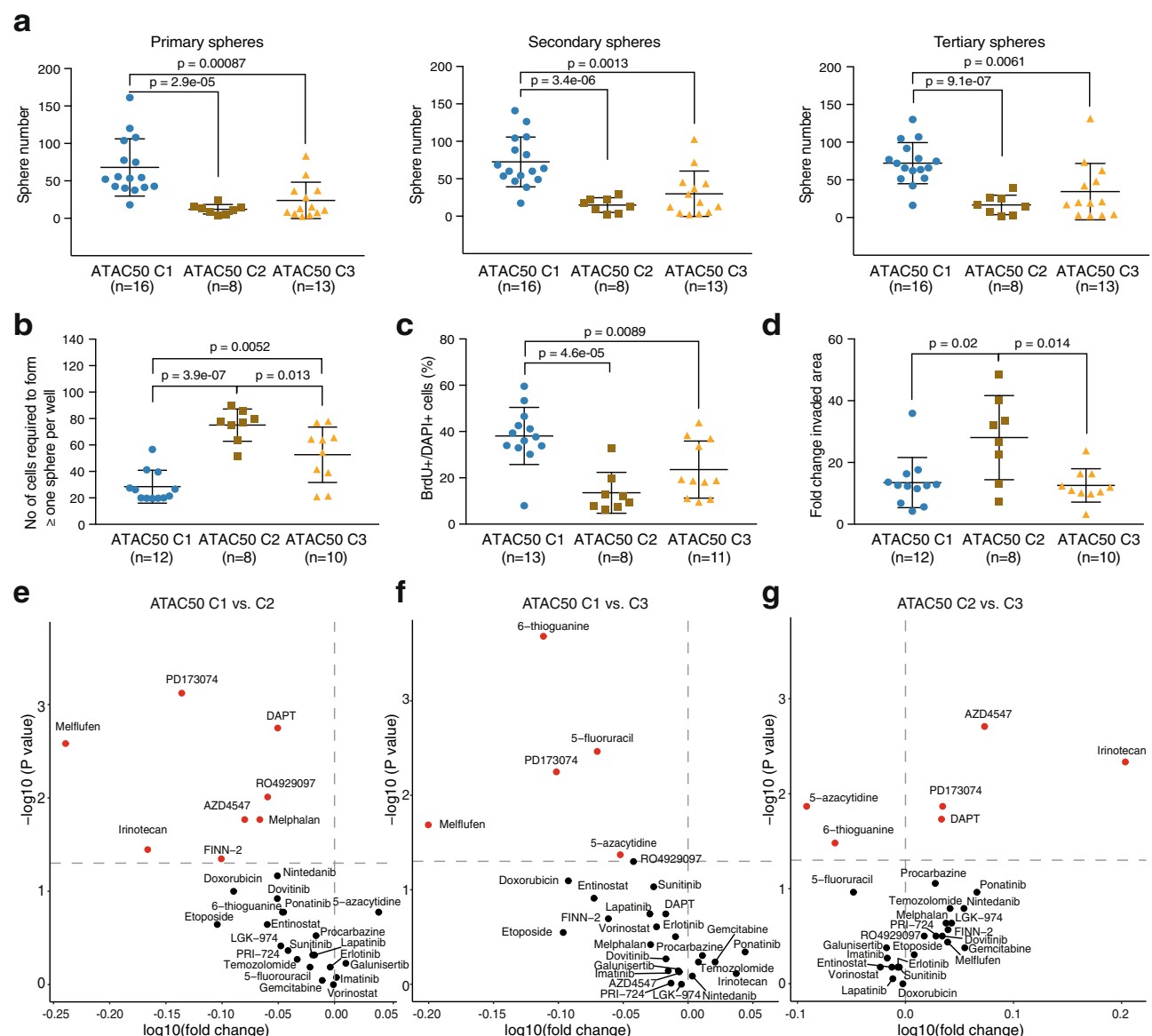

**Fig. 5 ATAC50 produces functional separation of hGSC cultures. a** Consecutive sphere assays comparing ATAC50 clusters. Primary spheres (left), secondary spheres (middle), and tertiary spheres (right). **b** ELDA comparing ATAC50 clusters. **c** Frequency of BrdU-positive cells. Data show mean ± SEM. **d** Collagen invasion assay measuring the invaded area at 24 h. Data show mean ± SEM. **a–d** Data show mean ± SEM. Two-sided Welch's t-test was performed on all pair-wise comparisons, and significant differences are indicated with p value. n number of cell cultures. **e–g** Volcano plots of pair-wise comparisons of AUC scores in ATAC50 C1 (n = 12), C2 (n = 7), and C3 (n = 10) cultures to 28 anticancer drugs. Red circles indicate significantly different drug response. n, number of cell cultures. Two-sided Mann–Whitney was performed on all pair-wise comparisons without multiple adjustment. **e** ATAC50 C1 versus C2. Red circles in upper left corner, C1 more sensitive. **f** ATAC50 C1 versus C3. Red circles in upper left corner, C1 more sensitive. **g** ATAC50 C2 versus C3. Red circles in upper left corner, C2 more sensitive. Red circles in upper right corner, C3 more sensitive. Source data are provided as a Source Data file for Fig. 5a–g.

**ATAC-based clusters can predict mouse and patient survival.** Orthotopic tumor growth is a defining capacity of cancer stem cells. We used in vivo survival data, in total 322 intracranially injected immune-deficient mice, from published[22,25,26,28] and unpublished experiments, and included only individuals that had been killed because of disease symptoms before the experimental endpoint (Supplementary Data 9). When grouping mice according to the ATAC50 clusters we found a significant difference in survival between all groups with C1 being most aggressive (Fig. 6a). The same data divided by TCGA subtypes showed significant differences between MS and CL or PN but could not separate all three groups (Fig. 6b).

We also analyzed survival of GBM patients from whom the hGSC cultures had been derived (Fig. 6c–f). When ATAC50 patients were grouped according to TCGA there were no survival differences (Fig. 6c), which is in line with previous patient data for IDH wt GBM. ATAC50 clusters produced more separated curves (Fig. 6d), although there were no significant differences. We also analyzed the MCO clusters because of the high degree of overlap with ATAC50 (Fig. 6e), which produced even more separated curves and a close to significant difference between MCO2 and MCO3 patients (Fig. 6e). Finally, we used all patients and the ATAC60 classification, which for the overlapping 50 hGSC cultures had produced essentially the same clusters as

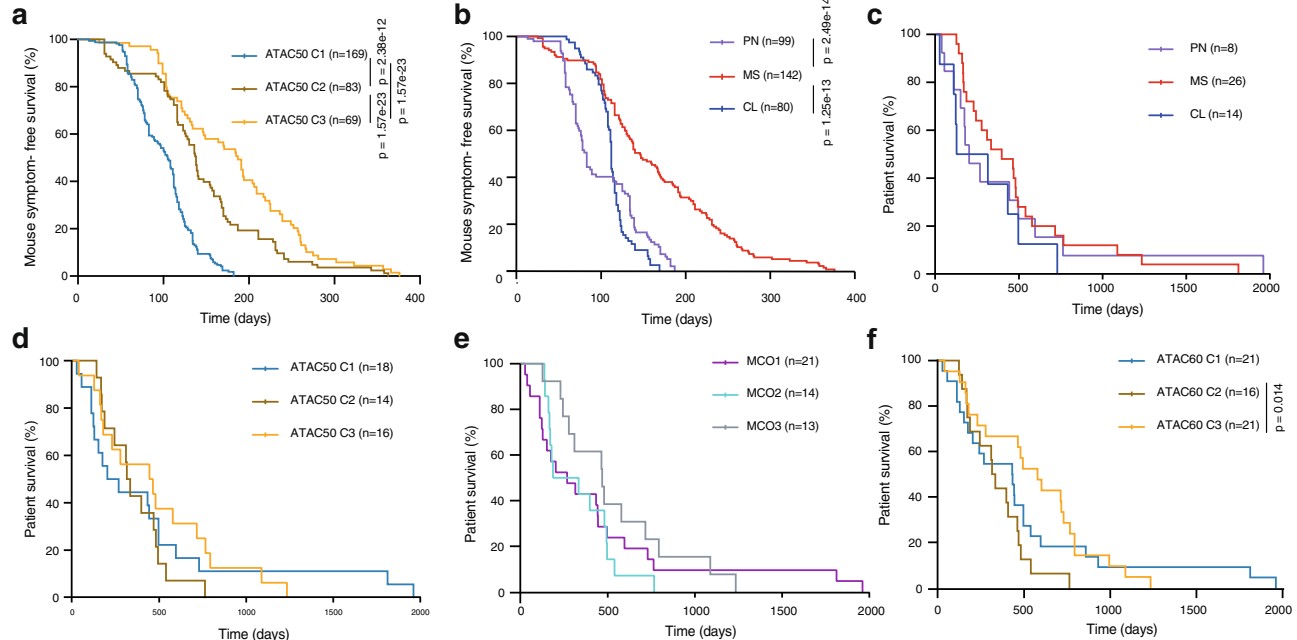

**Fig. 6 Survival analyses comparing ATAC50, TCGA subtype, MCO, and ATAC60 classifications. a**, **b** Kaplan–Meier analysis of symptom-free survival of immune-deficient mice intracranially injected with $10^4$, $10^5$, or $2 \times 10^5$ cells from 35 different hGSC cultures. $n$, number of injected mice. **a** Cultures are divided by ATAC50 clusters, C1 ($n = 14$), C2 ($n = 11$), C3 ($n = 9$). **b** Cultures are divided by TCGA subtypes, PN ($n = 6$), MS ($n = 21$), CL ($n = 7$). **c**–**f** Kaplan–Meier analysis of GBM patient survival. $n$, number of patients. **c** Patients are divided based on TCGA subtype. **d** Patients are divided based on ATAC50. **e** Patients are divided based on MCO. **f** Patients are divided based on ATAC60 clusters. C2 versus C3. Log-rank (Mantel–Cox) test was used and significant differences are indicated with $p$ values. Source data are provided as a Source Data file for Fig. 6a–f.

ATAC50 (Supplementary Fig. 3d). By adding ten more patients the curves between C2 and C3 patients became clearly significant and showed improved survival for C3 patients (Fig. 6f). Because of the clinically relevant ATAC60 patient stratification we reanalyzed the results of Figs. 5 and 6a using ATAC60 groups (Supplementary Fig. 7). This showed overall the same result as with ATAC50 with small changes in significance levels for proliferation (Supplementary Fig. 7c, slightly decreased between C1 and C2), invasion (Supplementary Fig. 7d, slightly increased between C2 and C1 or C3), drug response (Supplementary Fig. 7f–g, slightly decreased between C3 and C1 or C2), and in vivo tumorigenicity (Supplementary Fig. 7h, slightly increased between C2 and C3) which supports the robustness of the ATAC clustering.

In conclusion, TCGA subtypes had no predictive value while both the MCO- and ATAC-classifications could separate patients of the two most molecularly and functionally similar GBM subgroups. The substantial overlap of these classifications suggests that cell lineage-based classifications could be valuable to predict GBM patient survival, likely because of their ability to distinguish essential tumor cell phenotypes.

**Cross-species analyses reveal MCO prediction of hGSC ATAC clusters**. By a glance the MCO groups and ATAC50 clusters had shown certain similarities such as a PN-like phenotype of mGC1$_{GFAP}$ and ATAC50 C1, and a MS-like phenotype of mGC2$_{NES}$ and ATAC50 C2. To investigate this further we performed a cross-species comparison of enriched TF motifs in unique ATAC peaks (Supplementary Data 3 and 6) of both species (Supplementary Fig. 8a–c). We analyzed the positive correlation of the significantly enriched and variable TF motifs of mGC1$_{GFAP}$ (Supplementary Fig. 8a), mGC2$_{NES}$ (Supplementary Fig. 8b), and mGC3$_{CNP}$ (Supplementary Fig. 8c) separately with the complete list of enriched TF motifs from the ATAC50 data.

This showed that for mGC1$_{GFAP}$ the majority of positively correlating TF motifs were with C1 (OCT2, OCT4, OCT6, RFX2, RFX3). For mGC2$_{NES}$ the strongest correlation were with TF motifs enriched in C2 (AP-1, AP-2γ, AP-2α, ATF3, BACH1, BACH2, BATF, ERG, ETS1, FOSL2, JUNB, NFE2L2, NF-E2, NRF2), although there were a number of TFs that correlated with C1-enriched TFs (ERG, ETS1, HIC1, LHX1, NF1, SIX1, SIX2, SIX4). For mGC3$_{CNP}$ there were fewer overlapping TFs which was expected because of the intermediate phenotype of this group, and those that were positively correlating were enriched in C1 (HIC1, NF1, OCT4, OCT6) and C3 (AP-2a, TEAD2, TEAD3, PU.1). This supported the perception of a relation between mGC1$_{GFAP}$ and C1, mGC2$_{NES}$ and C2, and mGC3$_{CNP}$ and C3 and strengthened the connection between developmental origin and ATAC50 clusters.

Finally, we analyzed the relation of the cell lineage-based MCO stratification with ATAC50. We had already observed a considerable overlap of ATAC50 and MCO (Fig. 7a) and wanted to investigate the underlying reason. We started by analyzing if the chromatin landscape of the MCO genes could guide the ATAC50 clusters. Of the 196 MCO genes[22] we used 166 human homologs for which the ATAC-seq peaks of promoter regions were extracted and analyzed by NMF (Supplementary Fig. 8d, e). This produced a poor overlap with the ATAC50 clusters (Fig. 7b), consistent with the importance of DRE regions (Fig. 3h). Next we used the DRE regions of the MCO genes through the peak-to-gene linking prediction analysis (Supplementary Data 5). The MCO human homolog genes were annotated to 786 ATAC peaks of DRE regions that were analyzed by NMF (Supplementary Fig. 8f, g). This showed a higher concordance with ATAC50 (Fig. 7c) compared to promoter-guided clusters (Fig. 7b) but still lower than the MCO stratification (Fig. 7a). Then we used the cell of origin-specific mouse ATAC peaks ($n = 2075$, Fig. 1e and Supplementary Data 2) that were annotated to 2028 mouse genes, converted to 1629 human homolog genes of which 805 were

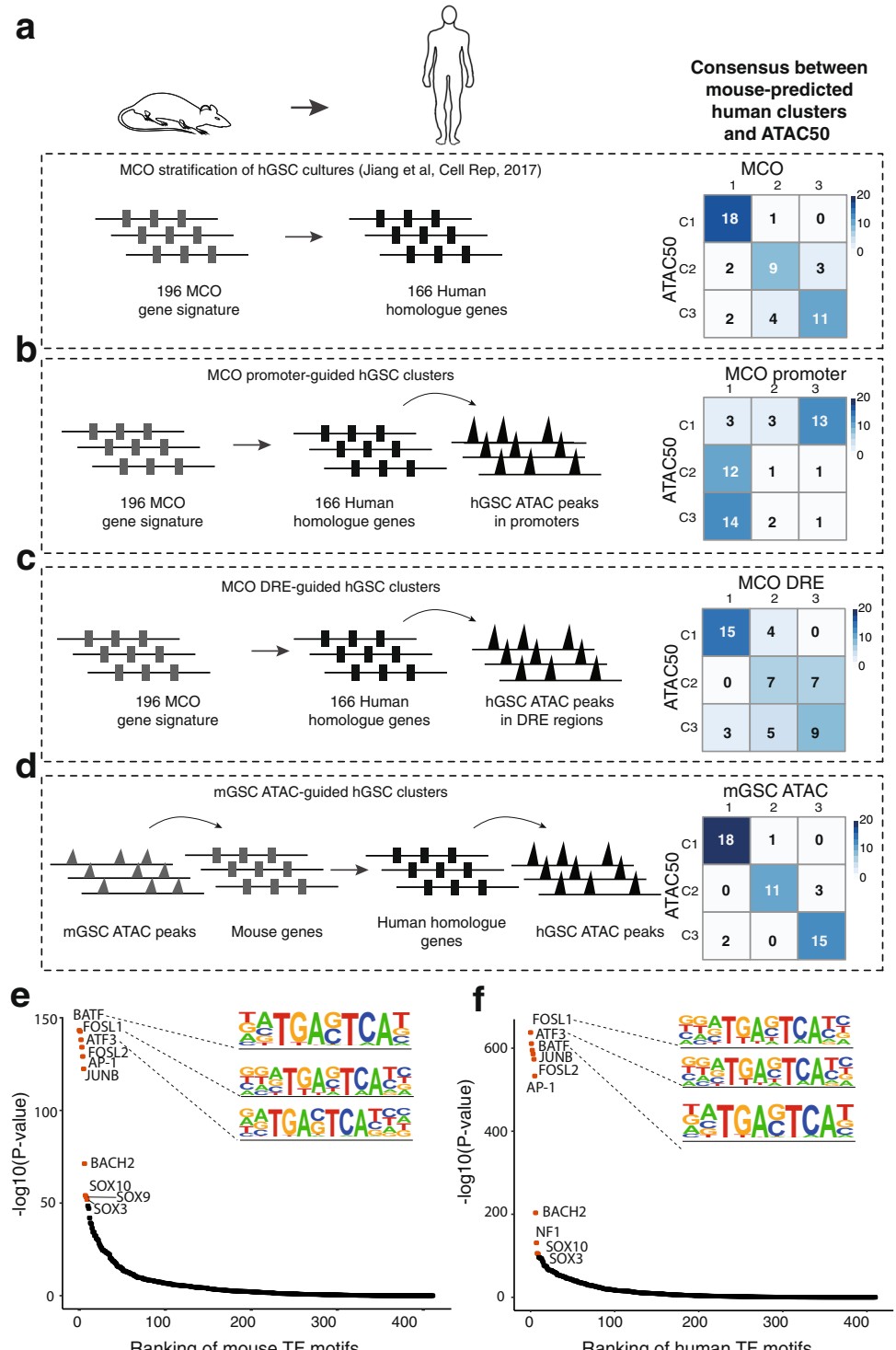

**Fig. 7 Cross-species analysis reveal conservation between mouse and human GSC chromatin accessibility. a–d** Analyses of mouse GSC-guided clustering of hGSC cultures and their overlap with ATAC50 clusters. **a** Overlap with MCO gene expression classification[22]. Schematic figures were produced by the authors. **b** Overlap with clusters produced by hGSC ATAC data at MCO gene promoters. **c** Overlap with clusters produced by hGSC ATAC data at MCO DRE regions. **d** Overlap with clusters produced by converting mGSC cluster-unique ATAC data to hGSC ATAC data through peaks-to-genes and genes-to-peaks conversions. **e** Ranking of enriched TF motifs in mGSC cluster-specific ATAC peaks. Red circles indicate top-10 significant motifs. **f** Ranking of enriched TF motifs in the hGSC ATAC peaks obtained in (**d**). Red circles indicate top-10 significant motifs. Source data are provided as a Source Data file for Fig. 7a–f.

present in the peaks-to-genes data and could be linked to 4834 human ATAC peaks. These peaks were used in NMF (Supplementary Fig. 8h, i) which produced clusters that, surprisingly, showed a very high agreement with the ATAC50 clusters

(Fig. 7d). As reference we compared this to 1000 NMF analyses of at each time 4834 randomly selected ATAC peaks from the peaks-to-genes data, which did not reproduce the ATAC50 clusters (Supplementary Fig. 8j). To investigate the regulation of

the open chromatin in the mouse and human ATAC data used in the mouse-dictated human ATAC50 clusters (Fig. 7d), we compared enriched TF motifs in the 2075 mouse ATAC peaks to those of the corresponding 4834 human ATAC peaks. This showed an 90% overlap of the top-10 significantly enriched TF motifs (Fig. 7e, f). As a reference, we compared this to a 1000 times repeated control experiment where enriched TF motifs in 2075 randomly selected mouse ATAC peaks and their corresponding human ATAC peaks were compared (Supplementary Fig. 8k). The TF motif overlap extended from 0 to 7 with an average overlap of 18.3% (standard deviation = ±18.7%).

Our cross-species analyses supported a molecular relation of the mGSC groups with the ATAC50 clusters, where the core set of overlapping TF motifs in differential mouse and human ATAC peaks showed a connection between mGC1$_{GFAP}$ and ATAC50 C1, mGC2$_{NES}$ and ATAC50 C2, and the intermediate characteristics of mGC3$_{CNP}$ and ATAC50 C3 (Supplementary Fig. 8a). The predictive capacity of the differential mouse ATAC peaks to produce almost identical hGSC clusters as ATAC50 (Fig. 7d) that displayed common TF regulatory mechanisms with the mGSC groups (Fig. 7e, f) supported the role of neurodevelopmental processes in GBM. In all, this provides strong support for the presence of a conserved cell of origin-determined epigenetic regulation of GBM.

## Discussion
Cross-species analyses of transcriptomes have been performed for many cancers, including primary malignant brain tumors such as medulloblastoma[40], ependymoma[41], and GBM[22,23], and have helped to extend our knowledge about how developmental mechanisms contribute to cancer. With increasing knowledge about the central role of epigenetic regulation in GBM and GSCs we set out to perform a cross-species analysis of chromatin-accessibility in mouse and human GSC cultures that we had previously analyzed by cross-species transcriptome analysis[22]. The contribution of developmental regulation in GBM biology and GSC function remains to a large extent to be deciphered and we reasoned that it would be a strength to use the same extensively characterized mouse and human GBM models to be able to compare the previous transcriptome data with the new genome-wide chromatin-accessibility data.

ATAC-seq analysis divided both mouse and human cells in three molecularly and functionally distinct groups, and for both species these groups were mainly determined by the chromatin state of DRE regions and regulated by contrasting sets of TFs. The three mGSC cell of origin groups were arranged, as we had already shown for transcriptome data[22], in a gradient of PN to MS phenotypes with mGC1$_{GFAP}$ and mGC2$_{NES}$ at each end and mGC3$_{CNP}$ in between. The three hGSC ATAC clusters aligned well with recent ATAC-seq analyses of human GBM cells and GSC cultures from independent cohorts[8,16] and supported our non-linear regression analysis which predicted our cohort to be large enough to capture the spectrum of GBM inter-patient heterogeneity. The hGSC ATAC clusters were similarly to the mouse cultures found along the PN to MS axis where expression of GSC state markers[7] and analyses of TF circuits showed that C1 cultures were glial stem and progenitor cell-like, C2 profoundly mesenchymal-like, and C3 intermediate with mostly astrocytic and mesenchymal traits. The cross-species analysis of enriched TF motifs in unique ATAC peaks of mouse and human data strengthened the connection between mGC1$_{GFAP}$ and ATAC50 C1, mGC2$_{NES}$ and ATAC50 C2, and mGC3$_{CNP}$ and ATAC50 C3. As previously shown for mGSC cultures of different origin[22], also ATAC50 clusters were functionally well-defined, where C1 cultures were most self-renewing, proliferative, tumorigenic, and

drug-sensitive, C2 cultures most invasive and drug-resistant, and C3 cultures least invasive and tumorigenic. Importantly, we found that the functional differences between C2 and C3 also were reflected in significantly different patient survival. The precision of chromatin accessibility to separate prognostic patient groups emphasized the importance of epigenetic regulation in GBM.

The considerable overlap of the MCO classification with the ATAC50 clusters implied a cell lineage-controlled regulation of hGSCs. This was corroborated by a second cross-species analysis where we used the unique chromatin-accessible regions of the mGSC data and converted those to chromatin-accessible regions of hGSCs which almost completely could re-establish the ATAC50 clusters. The 90% overlap of top-10 enriched TF motifs in the mouse and human accessible chromatin regions provided strong support for a conserved epigenetic cell lineage regulation of GBM. Our cross-species analyses also validated that our three PDGF-driven tv-a mouse GBM models can produce tumors representative of the breadth of developmental regulation present in our large collection of patient-derived GSC cultures. The fact that one oncogenic driver (PDGFRA activation) could reproduce the epigenetic heterogeneity of human GBM was in line with results from the comprehensive single cell RNA-seq analysis of human GBM[7] where multiple cellular states were shown to be present in all investigated tumors, while state distributions were proposed to be dictated by certain genetic factors such as PDGFRA. Taken together, this would argue for that GBM epigenetic heterogeneity is mainly the consequence cell of origin-inherited developmental regulation which in turn provide the basis for possible GSC states, where GBM driver mutations determine the state transition dynamics.

We show the power of a chromatin accessibility-based functional classification of GSCs. Continued work to identify the key regulatory elements in the DREs dictating the different properties and common features of the epigenetic clusters, and to validate key TF circuits regulating GSC states by perturbation strategies will be crucial to pinpoint therapeutic targets. Our analysis of chromatin accessibility in mGSCs and hGSCs has revealed a species conservation of the GBM epigenome and demonstrated the importance of cell lineage diversity for accurate in vivo modeling of GBM inter-patient heterogeneity.

## Methods
**Mouse GBM cell cultures**. Previously established mouse GSC and NSC cultures explanted from mouse primary GBM tissues or the SVZ of uninjected mice[22] were cultured on growth-factor depleted ECM-coated dishes (Sigma) in media containing DMEM/F12 GlutaMAX mixed 1:1 (GIBCO-Invitrogen) with addition of 1% penicillin G/streptomycin sulfate (Sigma), B-27 without vitamin A (1:50; GIBCO-Invitrogen), HEPES (0.2 mM; Sigma), and insulin (20 ng/ml; Sigma). Mouse GSCs were cultured without addition of growth factors while mouse NSC were cultured with FGF2 (20 ng/ml; PeproTech) and EGF (20 ng/ml; PeproTech). Mouse cells used in all experiments were below passage 13. All mouse cell cultures are listed in Supplementary Data 1.

**Human GBM cell cultures**. All 60 human GSC cultures used in this study have been established in our laboratory and are part of the HGCC biobank (hgcc.se). Handling of human tissues and data were performed in accordance with the protocol approved by Uppsala ethical review board (2007/353) and following informed written consent from all patients. All have been previously described and most have been authenticated[22,25,26]. Cultures were maintained on poly-ornithine/laminin-coated dishes in DMEM/F12 Glutamax (Gibco) and Neurobasal medium (Gibco) mixed 1:1 with addition of 1% B27 (Invitrogen), 0.5% N2 (Invitrogen), 1% penicillin/streptomycin (Sigma), 10 ng/ml each of EGF and FGF2 (Peprotech). Human cells used in all experiments were below passage 20. All human cell cultures are listed in Supplementary Data 1.

All cell cultures, both mouse and human, have been regularly analyzed for mycoplasma infection using either a PCR-based method with the primers Myco1 (50-GGCGAATGGGTGAGTAACACG) and Myco2 (50-CGGATAACGCTTGC GACTATG) (Invitrogen), or the KAPA kit (Techtum, cat# 25-KK7352), and have at all times tested negative.

**ATAC-seq of mouse and human cell cultures**. Omni-ATAC method was used on mouse GSCs as previous described[42]. In brief, mouse GSCs (mGC1$_{GFAP}$: SC81, SC83, SC84; mGC2$_{NES}$: SC50, SC52, SC64; mGC3$_{CNP}$: SC37, SC74, SC112) were counted and 50,000 cells were used per omni-ATAC reaction. Counted cells were centrifuged at 500$g$ in a fixed angle microfuge for 5 min at 4 °C and the supernatant was discarded. Cell pellets were resuspended in 50 μl resuspension buffer (10 mM Tris-Cl, pH 7.4, 10 mM NaCl, 3 mM MgCl$_2$, 0.1% NP40, 0.1% Tween-20, and 0.01% digitonin), and incubated on ice for 30 min. Then, 950 μl resuspension buffer was added and cells were centrifuged at 500$g$ in a fixed angle microfuge for 10 min at 4 °C. The supernatant was discarded, cell pellets were gently pipetted in transposase mixture (12.5 μl 2× transpose buffer, 16.5 μl 1× PBS, 0.5 μl 1% digitonin, 0.5 μl 10% Tween-20, 2.5 μl Tn5 transposase (2 μM), 5 μl nuclease-free water), and incubated 30 min at 37 °C. The transposase mixture was purified with MinElute PCR Purification kit (Qiagen, 28004) and eluted in 10 μl Qiagen EB elution buffer. Sequencing libraries were prepared following the original ATAC-seq protocol[43].

Human GSCs and mouse NSCs were fixed with 1% formaldehyde (Thermo Fisher Scientific, 28906) for 10 min and quenched with 0.125 M glycine for 5 min at room temperature. After the fixation, ATAC-seq was performed as previous described[44]. Cells were counted and 50,000 cells were used per ATAC-seq reaction. The transposition reaction followed the normal ATAC-seq protocol. After transposition, a reverse crosslink solution (final concentration 50 mM Tris-Cl PH 8.0 (Invitrogen, 15568-025), 1 mM EDTA (Invitrogen, AM9290G), 1% SDS (Invitrogen, 15553-035), 0.2 M NaCl (Invitrogen, AM9759) and 5 ng/μl proteinase K (Thermo Scientific, EO0491)) was added up to 200 μl. The mixture was incubated at 65 °C with 1200 rpm shaking in a heat block overnight, then purified with MinElute PCR Purification kit (Qiagen, 28004) and eluted in 10 μl Qiagen EB elution buffer. Sequencing libraries were prepared following the original ATAC-seq protocol[43]. All sequencing was performed on Illumina NovaSeq 6000, and at least 20 million paired-end sequencing reads were generated for each ATAC-seq library.

**ATAC-seq data processing and quality analyses**. All ATAC-seq data were processed with same pipeline described below. After the Adapter sequence trimming, the ATAC-seq sequencing reads were mapped to genome hg19 (for human GSCs) or mm9 (for mouse GSCs and mouse NSCs) using bowtie2 (ref. [45]). Mapped paired reads were corrected for the Tn5 cleavage position with shifting +4/−5 bp depending on the strand of reads. All mapped reads were extended to 50 bp centered by Tn5 offset. The PCR duplication were removed using Picard (http://broadinstitute.github.io/picard/) and sequencing reads from chromosome M were removed. The Peak calling of each ATAC-seq library was performed with MASC2[46] with parameters -f BED, -g hs, -q 0.01, -nomodel, -shift 0. Peaks were merged into matrix with bedtools merge[47]. Raw reads within peaks were normalized using EdgeR's cpm[48]. Log transformation were applied on these normalized peaks to calculate the pearson correlation among duplicates. Unique ATAC peaks for hGSC ATAC clusters were selected using DESeq2 (ref. [49]), with cutoff $p$ value <0.01, FDR < 0.01, log2 fold change > 1, peak average intensity > 30, and coefficient of variance < 0.2. Mouse differential ATAC-seq peaks were identified by comparing each mGSC group with each other using the parameters log2(fold change) > 1, false discovery rate < 0.05.

The ATAC-seq saturation analysis from human GSC was performed by randomly selecting samples and successively calculating the number of peaks identified within the number of samples. The self-starting non-linear mode has been used to predicate the saturation point.

Non-linear model:

$$P(x) = a + b * e^{cx+d}$$

where $P(x)$ represents predicted numbers of peaks, $x$ corresponds the actual number of peaks, $a$, $b$, c, and $d$ represent the parameters for self-starting simulation.

For ATAC-seq peak visualization, Washu Epigenome Browser was used to visualize these presentative peak regions from mouse NSC, mGSC and hGSC.

To analyze the chromatin accessibility signal per gene, the accessibility of the regions (no further than the window −1000 to +100 bp from a transcriptional start site) were defined as promoter regions and the elements (located more than 3 kbp from a TSS and no further than 500 kbp) were represented as DREs. The genomic annotation of ATAC-seq was performed with seven genomic features: 3′ UTR, 5′ UTR, exon, intergenic region, intron, TSS, and TTS using ChIPseeker[50].

For ATAC-seq peak visualization, Washu Epigenome Browser was used to visualize these presentative peak regions from mouse NSC, mGSC, and hGSC.

**Annotation of unique ATAC peaks to genes and GO analysis**. Genomic annotation of each ATAC peak to its nearest gene for mouse (Supplementary Data 2) and human (Supplementary Data 5) was done using ChIPseeker[50]. STRING (https://string-db.org/) was used to perform GO enrichment analysis and the result was ranked by strength.

**NMF cluster analysis of human GSC ATAC-seq data**. The NMF method[51] was used to cluster Human GSC ATAC-seq with nsNMF[52]. In brief, ATAC-seq peaks were ranked according to their variance from high to low. The cophenetic correlation score was calculated with cluster number 2 to 8, and used to determine the

number of clusters following the standard method[51]. Top 70000 (20%) ATAC-seq peaks from hGSC were used to build NMF clusters.

**Genomic segmentation analysis for the human GSC ATAC-seq data**. The chromatin-state discovery and genome annotation for the ATAC-seq peaks from the human GSC ATAC-seq peaks was performed with ChromHMM[53] by downloading the data from following dataset: GSE119755 (H3K27ac ChIP-seq); GSE121601 (H3K27ac and CTCF ChIP-seq); GSE92458 (H3K4me1 and H3K27ac ChIP-seq); GSE74557 (H3K27me3 and H3K4me3 ChIP-seq). In total, seven chromatin status referred to Epigenomic Roadmap Consortium were defined: active promoter (H3K27ac and H3K4me3 together), active region (H3K27ac alone), inactive regions (H3K27me3 alone), insulator (CTCF), strong enhancer (H3K4me1 and H3K27ac together), weak enhancer (H3K4me1 alone) and no signal, were used to characterize the human GSC ATAC-seq peaks.

**Cluster-specific TF motif enrichment analysis of unique ATAC-seq peaks**. Specific ATAC-seq peaks for different mGSC groups and different hGSC clusters were first calculated with DESeq2 using default parameters, and the specific differential peaks for each group/cluster were analyzed for TF motif enrichment. In brief, the Homer vertebrate TF database was used as input of TF motifs in chromVAR, then TF accessibility deviation values for each sample were calculated across the whole sample set. TF deviations with a threshold larger than 1 were kept, and TF motifs with positive correlation with one group/cluster was selected to represent that group/cluster. TFs were ranked based on their variabilities for each group/cluster, and $z$-scores of deviations from each TF were visualized in a heatmap.

**Correlation analysis of enriched TF motifs and corresponding TF gene expression**. To more accurately predict TF activity, TF deviation and gene expression were combined in a Pearson correlation analysis with the threshold $p$ value <0.05.

**Identification of significantly enriched and variable TF motifs of unique ATAC-seq peaks**. To identify all significantly enriched and variable TF motifs of the mGSC group-unique and hGSC cluster-unique ATAC peaks the deviation score of TF motifs was positively correlated to each cluster with the cut-off parameters of variability >1.5 and $q$ value <0.05. TFs representing each mGSC group are listed in Supplementary Data 3 and TFs representing each ATAC50 cluster are listed in Supplementary Data 6.

**Linkage of hGSC ATAC-seq data to genes and gene expression correlation analysis**. To analyze the correlation between human ATAC-seq data and gene expression array data we performed a correlation-based approach. First, ATAC-seq peaks were annotated to their nearest genes ("peak-to-gene linkage"; Supplementary Data 5) within ±0.5 Mbp but ±3 kbp of TSS. For each pair, the Pearson correlation between the ATAC-seq peak accessibility and the gene expression level was calculated. Next, the mean and standard deviation for these correlations were calculated to represent nonspecific correlation. Then, multiple correction was performed using Benjamini–Hochberg procedure to adjust these $p$ values. At last, only pairs with false discovery rate (FDR) < 0.05 were kept.

**TFs footprint analysis for cluster-specific hGSC ATAC-seq peaks**. In previous descriptions, Tn5 transposase inserted two adaptors separated by 9 bp[54]. Sequencing reads aligned files in sam format by offsetting +4/−5 bp for all the reads depending on the strand of reads. A shifted base sam file converted to bam format and was sorted by samtools[55]. ATAC-seq reads for each ATAC50 cluster of samples (C1, C2, and C3) were concatenated and 200 million reads were randomly selected from each cluster and merged into bam files. Then TF footprint analysis was performed on cluster-specific regions using the HINT-ATAC software. Input motifs were obtained from the Homer[56] database of vertebrates. Four hundred and fourteen motifs were tested and filtered with $p$ value <0.05. The normalized read counts were centered by the motif sites around 200 bp genomic region for visualizing motif footprints.

**Consecutive sphere formation assays**. Adherent human GSC cultures were dissociated with TrypLE (Thermo Fisher, 12563011) into single-cell suspensions. For primary sphere formation, 1000 cells/well were seeded in eight replicates in a 24-well low attachment plates. After 7 days, the number of primary spheres formed for each culture were counted. The primary spheres were dissociated and 1000 cells/well were again seeded in eight replicates for secondary sphere formation that were counted after 7 days. The same procedure was repeated for the tertiary spheres.

**Proliferation analysis**. Human GSC cultures ($5 \times 10^3$ cells/well) were seeded (Day 0) on laminin-coated coverslips in a 24-well plate using serum-free medium. The next day (Day 1) 1 μg/μl of BrdU (Sigma, B5002) was added to each well for 16 h before they were fixed with 4% formaldehyde (Histolab, 02176). After fixation cells

were washed with PBS and permeabilized in 2 M HCl for 20 min, followed by the washing again with PBS. Cells on the coverslips were permeabilised in 0.2% Triton X-100 with 3% bovine serum albumin (Sigma, A7906) for 5 min and washed thrice with PBS. Cells were blocked in 0.2% Triton X-100 solution containing 1% BSA and 5% normal goat serum (Dako, X0907) for 1 h. Primary antibody again BrdU (1:100; Abcam, Ab6326) was applied overnight in a humidified chamber at 4 °C. Cells were washed three times with 0.2% Triton X-100 solution for about 5 min each time. Secondary antibody incubation was performed for 30–60 min in room temperature with anti-rat Alexa 555 (1:400; Invitrogen, A21434). Lastly cells were washed three times with 0.2% Triton X-100 solution for about 5 min each time and mounted in Fluoromount (Sigma, F4680) with 0.1% DAPI in it. The stainings were visualized and quantified under a LEICA DMi8 Fluorescent microscope. The experiment was repeated three times on consecutive passages for all the cultures.

**Extreme limiting dilution assay**. Human GSC cultures were dissociated into single-cell suspensions in serum-free medium. Cells were seeded in a 96-well low attachments plates (CLS3474-24EA; Sigma) with the seeding density ranging from 100 cells to 1 cell per well, with 10 replicates per condition. After 7–10 days, the number of wells without spheres for each cell density were counted. The number of cells required to form at least one sphere per well was calculated by extrapolating the values of $x$-intercept for each culture and plotted using PRISM 7 software. The experiment was repeated three times for all cultures.

**Invasion assay**. Cell spheres obtained from hGSC cultures by seeding 50 or 100 cells per well in the ELDA experiment were used to measure the invasive capacity for each culture. Collagen gel matrix was prepared and spheres were transferred into the collagen gel matrix sandwich in a 24-well plate as described previously[26]. Pictures were taken after 10 min and 24 h in an Eclipse TS 100 Nikon microscope. Image J 1.52a software was used to measure the invasion area for each sphere. From each hGSC culture at least 10 spheres were analyzed. The experiment was repeated three times on consecutive passages for all the cultures.

**Drug response analysis in HGCCs**. A panel of 28 anticancer compounds (Supplementary Data 8) were used to measure the drug sensitivity of hGSC cultures. Cells were seeded, 1000 cells/well in a poly-ornithine (Sigma, P3655) and laminin (Sigma, L2020) coated 384-well plate (Thermo Fisher Scientific, 164688) with six replicates for each culture and dose. The following day cells were exposed to the compounds and drug response was measured after 72 h with the non-clonogenic fluorometric microculture cytotoxicity assay. The experiment was repeated twice. Dose response curves were plotted and the average area under the curve values for each compound and culture were calculated. For each ATAC50 cluster (C1, C2, C3) the average of AUC values for each compound was calculated and compared, pairwise, between ATAC50 clusters. The log10 fold change between clusters were calculated using Wilcoxon test and scatter plots produced as described previously[21].

**In vivo xenograft analysis**. All animal experiments were performed in accordance with the rules and regulations of Uppsala University and approved by Uppsala animal ethics committee (C237/12 and C182/14). Intracranial cell transplantations of human GSC cultures were performed in neonatal *NOD.CB17-Prkdcscid/NCrHsd* mice (Harlan) of both sexes as previously described[22,25,26,28]. Of the 321 mice included in the study 60 were previously unpublished (Supplementary Data 9). In brief cells were dissociated in TrypLE and resuspended in DMEM/F12 medium. A volume of 2 μl cell suspension with cells ranging from 10,000 till 200,000 was orthotopically injected using a motorized stereotaxic injector (Stoelting CO). The coordinates measured from lambda were anterio-posterior 1.5 mm, medio-lateral 0.7 mm, and dorso-ventral 1.5 mm. Mice were monitored every second day and euthanized through exposure to carbon dioxide upon symptoms of illness according to the ethical permit humane endpoint, determined by Uppsala University guidelines on the recognition of pain, distress, and discomfort in experimental animals. Such symptoms could for example include immobility, piloerection, hunched posture and weight loss of 10%. Only mice that showed disease symptoms before the endpoint of the experiment were used in the survival analysis.

**Quantification and statistical analysis**. Statistical analysis was performed using GraphPad PRISM 7 software or R version 3.4.0. Figures containing data from multiple repetitions of experiments were presented as mean ± SEM. For sphere-formation, ELDA, proliferation, and invasion experiments Student's $t$-tests were performed. For mice and patient survival, log-rank (Mantel–Cox) test was the statistical method used to calculate the significance in between the groups/clusters.

**Cross-species analysis of TF motifs in cluster-specific mouse and human ATAC peaks**. The deviation scores of significantly enriched and variable TF motifs of mouse (Supplementary Data 3) and human (Supplementary Data 6) GSC cultures were compared. If the average deviation was larger than zero the TF was defined as positively correlated.

**Cross-species analyses of MCO genes and unique mouse ATAC peaks**. MCO genes were converted to human homolog genes and ATAC-seq data of promoter regions or DRE regions of these genes were used to build NMF clusters. For the mGSC ATAC-guided clusters we used the annotated mouse genes from the unique mouse ATAC peaks (Supplementary Data 2) and converted these to human homologs. The human homolog genes were converted to human ATAC peaks through the peaks-to-genes analysis (Supplementary Data 5). NMF analysis was performed on these hGSC ATAC peaks. To analyze the major regulatory sequences in the (input) unique mGSC ATAC peaks and the (output) hGSC ATAC peaks we performed TF motif enrichment analysis using HOMER of both data sets and compared the top-10 ranked enriched TF motifs.

**Reporting summary**. Further information on research design is available in the Nature Research Reporting Summary linked to this article.

## Data availability

All raw data ATAC-seq data from human GSC and mouse GSC generated in this study have been deposited in Gene Expression Omnibus of National Center for Biotechnology information under access code GSE163853. The publicly available gene expression data of human GSC used in this study is from previous published report[22] and available in the Gene Expression Omnibus of National Center for Biotechnology information under access code GSE91393. The in vivo survival data of intracranially injected immune-deficient mice from previous published reports[22,25,26,28] are available in Supplementary data 9. The remaining data are available within the Article, Supplementary Information or Source Data file. Source data is provided with this paper.

## Code availability

All detailed scripts used in this study were deposited and can be accessed via the link https://github.com/chenlab2019/GSC and the corresponding DOI is as follows: https://doi.org/10.5281/zenodo.6374965 (ref. [57]).

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

## Acknowledgments

We thank the HGCC and U-CAN biobanks at Uppsala University for providing human material, and members of the Chen and Uhrbom laboratories for scientific discussions. Sequencing was performed by the SNP&SEQ Technology Platform in Uppsala. The facility is part of the National Genomics Infrastructure (NGI) Sweden and Science for Life Laboratory. The SNP&SEQ Platform is also supported by the Swedish Research Council and the Knut and Alice Wallenberg Foundation. This work was supported by the Swedish Research Council (2016-06794, 2017-02074 to X.C., 2018-02906 to L.U.), the Swedish Cancer Society (15 0877, 18 0763, 21 1518 to L.U., 21 1449, 22 0491 JIA to X.C.), Beijer Foundation (to X.C.), Jeansson's Foundation (to X.C.), Petrus och Augusta Hedlunds Stiftelse (to X.C.), Göran Gustafsson's prize to younger researchers (to X.C.), Åke Wibergs Stiftelse (to X.C.), Vleugel Foundation (to X.C.), and Uppsala University (to X.C.).

## Author contributions

L.U. and X.C. conceptualized, designed, and supervised the study. N.P.M. planned, performed, and analyzed most experiments. M.J., I.Y., L.Z., Y.D., Y.X., E-J.T., and X.C. performed additional experiments. X.L. performed all computational analyses with help from P.X. R.L. and M.F. provided technical and material support. L.U. and X.C. wrote and edited the paper with input from all co-authors.

## Funding

## Competing interests

R.L. and M.F. are minor shareholders of Oncopeptides AB and have previously received unrestricted grants from the company. The remaining authors declare no competing interests.
