## [Peer Review File · Nature Communications]

Cell-lineage controlled epigenetic regulation in glioblastoma stem cells determines functionally distinct subgroups and predicts patient survivalReviewers' Comments:

Reviewer #2:

Remarks to the Author:

In the manuscript by Lu et al. the authors performed a cross-species epigenome analysis of mouse and human GSC cultures. The authors analyzed and compared the chromatin-accessibility landscape of nine mouse GSC cultures of defined cell of origin and used sixty patient-derived GSC cultures by assay for transposase-accessible chromatin using sequencing (ATAC-seq). The authors have claimed to have uncovered a variability of both mouse and human GSC cultures that was different from transcriptome analysis and better at predicting functional subgroups.

Major Criticism:

1) Starting with the ATAC method, they used formaldehyde to fix it and then reverse crosslink it after tagmentation which is something the communicating author used in a previous ATAC-seq paper. For ATAC-seq I believe the need to fix the cell due to fluorescent staining and visualization afterward, but here they are only using those tissue for bulk ATAC, therefore, what's the precise reasoning for doing the formaldehyde crosslinking. Even if the authors claim using their method wouldn't change the ATAC profile, most in the field still prefer the option of capturing the chromatin in its native state without any crosslinking. Then why is there any need to use this method? It seems to create problems and raises concerns.

2) This reader doesn't really understand the need for including the mouse data as aspect of the study for Figure 1 and should remove this to strengthen the authors' rationales. This data appears to detract from their observations in the first figure of the manuscript. The authors used cells from three different origins (used in triplicate representing 9 cell lines) and performed bulk ATAC. One would predict that cells from different origins should definitely cluster well together, and what was presented in Supplementary Fig. 1C basically represents a problem using the triplicate or poor-quality reads. While it is totally understandable since it's from different mice and only 3 in each group can result in huge variation observed. The authors should either remove the outliers and increase the number in each group to see if the PCA clustering improves.

Figure as Supp1B basically shows no significant clustering between those SCXX samples by Pearson correlation, but they switch to NMF method to make it look more significant in Figure 1d. In Figure 1C, the authors argue the peaks are specific to different origins, but in fact they are also trying to split replicates of each origin and mix them in groupings based on the NMF clustering.

Through Figures 1E and 1F is basically regarded as hacking the statistic analysis to fit their proposed model. A knowledgeable reader would say they could see a clear trend in Figure 1F between those different origins, even though they are not significant in p values, as well as shown in Figure 1E, the authors just switched some of the samples to fit it as statistically significant even though these two figures resemble each other. This is a problem in the reader's view.

In Figure 1G the authors only displayed their new grouping to show some survival differences. This then questions how does one explain about the difference survival analysis of the origins then? That too is also likely different and maybe more significant than the authors' original intent.

Found in Figure 1G are TFs the authors selected from Figure 1i, which is presumed to yield a difference in binding judging from the heatmap, but their plot showed quite the opposite. Most knowledgeable readers would not regard these binding differently as this is acknowledged merely by the insignificant p values.

3) For addressing the human analysis, the number of samples used and data itself are truly valuable, and capturing brain tissue is not an easy task, so there is strong appreciation for what was needed for

them to generate those datasets, the QC looks OK, but this too still widely varies between samples used. This is understandable since patient variability and their collecting at different times by different personnel may represent the inconsistencies in some of the QC standards. The variation in TSS enrichment and Frip scores would lead to the wide variation in percent of promoter and enhancer peaks, where this is likely due to their technical inconsistencies with this difficult tissue type, but this aspect should not be listed as a biological difference and the focus in Figure 3. (Figure 3, panels f,g,h doesn't provide much real information). This questions the value in using this information. Figure 4, panels a,c,d is still question those distribution differences by ATAC peaks shown, even though called "significant" there is not much difference by eye, and this too could be due to technical variation. The authors used the same NMF clustering 50 human samples which doesn't correlate well with TCGA subtypes and mildly correlate with MCO scoring. What the authors claimed is their ATAC is yields a better predictive value and can capture sample difference that can't be found in the RNA-seq results, this is not believable based on the data presented thus far. From what we know, RNA-seq has less technical variation and should be more robust to present sample differences. Whatever marks survival, whether through sphere growth or differences in drug response, this reader believes you can find a similar trend using any other random grouping method that wouldn't carry any significant biological meaning. The way the authors are trying data mining with the data shown that doesn't not cluster well with real subtypes is problematic. If their 50 patients NMF clusters matches poorly with TCGA data, then this too is a problem. The authors are effectively claiming their datasets are better predictor than all the TCGA data combined. If by comparison the authors would need much more experimental and clinical data to back up their claims for the groupings shown. A reader would have argued that the authors should have focused more on the MCO score correlation and strengthen a story around that concept, where at the very minimum this would at least show readers that ATAC-seq data can be used to evaluate/correlate with MCO and find strong biological relationships more effectively, at least some of the TF activity in Figure 4h does look difference, or do a integrate RNA and ATAC analysis using MCO grouped RNA-data with their ATAC data and see how the different MCO group glioblastoma regulome works, that would be more interesting than those promoter and enhancer percentage QC and correlation analysis.

Overall evaluation is that the authors would at the very least need a major revision and to rethink about what and how they want to analyze this truly valuable dataset. What they are claiming now remains arbitrary and is weak without further biological and clinical data to back up their findings and observations.

Reviewer #3:

Remarks to the Author:

In the manuscript 'Conserved cell-lineage controlled epigenetic regulation in human and mouse glioblastoma stem cells determines functionally distinct subgroups', Lu et al. describe a novel approach for classifying GSC cultures based on transposase-accessible chromatin distribution. They identified three independent ATAC-seq clusters in 9 murine and 60 human GSC cultures that were different compared to gene expression-driven approaches. In their mouse model, they found functional differences in sphere-forming capacity, in vivo aggressiveness, drug response, and transcription factor motifs in the ATACs. They further went on performing similar experiments with human GSC cultures. The sequences were predominantly DREs in both species. Also, in the human system, they report functional differences between the clusters in terms of GSC clonogenicity and drug response. The human ATAC-seq clusters overlapped with the MCO stratification better when compared to the murine GSCs. Importantly, individuals belonging to two clusters showed significantly different survival in a small cohort of patients. Finally, they analyzed the TF motifs in the murine and human GSC ATAC-seq samples and found a significant overlap of these motifs.

The manuscript is well written and describes an in-depth analysis of ATAC-seq data obtained from a big collection of GSC cultures and shows that the patients can be clearly divided into well-defined clusters with this method. However, it is a retrospective study with few biologically functional data. Thus, the practical value of this approach in the classification of GB over the current gene expression-driven methods is not entirely evident.

Major points:

1) In the mouse model (figure 1), the authors use ATAC-seq to divide the GSC cultures into three distinct and clearly separated clusters. Then they use this scheme to perform their first functional tests (e-h) to later unite clusters A and B in Figures i and j. This seems a bit forced. If the authors suggest ATAC-seq as the basis for grouping the GSC cultures and three clear ATAC-seq clusters were identified, the differences within these three clusters should be analyzed. Given the investigation's retrospective nature, this grouping should not be changed depending on the experimental question to be answered.

2) In the sphere-forming assay (figure 1e and f), the behavior of the murine GSC cultures grouped by MCO and ATAC-seq are compared. The analyses show only differences for the ATAC-seq clusters. The authors describe that the system of murine GSCs was published before (Cell Reports 18, 977–990, January 24, 2017). How can it be explained that a similar analysis done in the cited paper showed highly significant differences between mGC1(GFAP) and mGC2(NES) or mGC3(CNP), while the new analyses did not?

3) Critics concerning data selection apply to the data obtained for the human GSC samples. In figure 2, the authors describe the identification of robust clusters with 60 GSC samples (ATAC60). It is not entirely clear why they remove 10 samples for further analyses. In the text, this procedure is explained by the lack of genetic data for the 10 lacking samples but with such a strong bioinformatic background these analyses should be possible and done.

4) While the xenograft data in figure 6 is convincingly showing that mice injected with human GSCs belonging to different ATAC-seq clusters have different survival rates, the patient data is not so strong. This may be at least partially due to the low number of patients. It may be helpful to include the patients for the ATAC60 panel to strengthen the significance (this could even be done without further information concerning the gene expression profiling). Especially because the different survival between two clusters is also underlined in the abstract.

5) The reviewer has not fully understood the discrepancy of the temozolomide sensitivity measured in Figures 6f and 5 e-g. If these differences are too subtle to be detected in AUC analyses, it may be enough to explain the discrepancies for the survival in the mouse model and the patients. The authors may test their hypothesis in vivo for some of their cultures to better mimic the patients' situation.

6) The finding that murine and human GSC ATAC clusters are largely overlapping is very interesting but may be biased by the growth conditions. The GSC cultures are established and kept in defined media with dominant growth factors (FGF2 and EGF). These could lead to the activation of selected signaling pathways in both murine and human GSCs that, in turn, may result in the activation of similar transcription factor patterns. This should be investigated carefully because TFs of the AP1 family are downstream substrates of MAP kinases that are known to be strongly activated by growth factor signaling. Thus, the authors may control if the culture conditions influence the ATAC-seq data.

Minor points:

1) From the M+M part, it is not clear how the sphere formation assay was done. Were the primary spheres established from in vitro cultures or from freshly dissociated xenografts? If they were established from in vitro cultures (which could explain the identical rates of primary, secondary, and tertiary spheres), it is a bit misleading to describe them as primary spheres. In this case, it would be better to rate it as in vitro proliferation.

2) The same question applies to the ELDA and the proliferation analyses. Were the cells from freshly dissociated tumors or from GSC in vitro cultures?

3) For the survival analyses in figure 6a, the mice were inoculated with different numbers of cells. Although this becomes evident from the supplementary table, it should be clearly stated in the text.

**Overview:**

We are grateful to the reviewers for taking their time to evaluate our work and providing
constructive comments and valid critique. We have found them very helpful to identify
weaknesses and unclarities of our manuscript and have considered them carefully. Below
you can find, point-by-point how we have addressed and responded to them. By doing so we
think that the revised manuscript has been greatly improved.

The main changes in the revised manuscript are:

**i)** A major concern raised by both reviewers was that the mouse GSC (mGSC) ATAC-seq
clusters did not produce cell of origin groups. Reviewer 2 suggested that this could have
been a consequence of the ATAC-seq data quality. To prove that comment right or wrong we
re-analyzed the nine mGSC cultures with Omni-ATAC-seq (*Corces R et al, Nat Methods,*
*2017, PMID: 28846090*). This method has been shown to enhance the signal-to-background
ratio and produce high-quality chromatin-accessibility data. The new mGSC ATAC-seq data
was of substantially higher quality and lower variability compared with the previous mGSC
data, as shown below in **Response 2.3** and **Response fig. 1**, and when we analyzed the
new data with PCA we found that mGSC cultures of the same cell of origin clustered
together, in alignment with the cluster analysis of transcriptome data of the same mGSC
cultures (*Jiang Y et al, Cell Rep, 2017, PMID: 28122246*). We have exchanged the previous
mouse ATAC-seq data with the new Omni-ATAC-seq data in the revised version of the
manuscript, and have re-analyzed all figures affected by that (**Fig. 1; Fig. S1; Fig. 7; Fig.**
**S7**). Importantly, we have not done this because the new data generated cell of origin
clusters but because of its substantially higher quality.

**ii)** The new mGSC groups showed a higher cross-species molecular connection with the
human GSC (hGSC) ATAC50 clusters. We had previously shown that the mGSC
transcriptome groups clustered with proneural (mGC1_{GFAP}, mGC3_{CNP}) or mesenchymal
(mGC2_{NES}) glioblastoma (*Jiang Y et al, Cell Rep, 2017*). This was now corroborated by
analyses of genes (**Fig. 1e**) and TF motifs enrichments (**Fig. 1g; Fig. S1g-i**) of the unique
ATAC peaks for each mGSC group, and a new cross-species analysis of enriched TF motifs
in unique mouse and human ATAC peaks showed a positive correlation between mGC1_{GFAP}-
C1, mGC3_{CNP}-C3 and mGC2_{NES}-C2 (**Fig. S7a**), which in all supported an alignment along a
proneural (mGC1_{GFAP}) to mesenchymal (mGC2_{NES}) axis with mGC3_{CNP} in between.

**iii)** A comment from reviewer 2 made us realize that we did not show the functional data of
hGSC cultures (**Fig. 5; Fig. 6a**) based on TCGA groups in the previous manuscript, which
we agree would be a relevant comparison to our ATAC50 groups. As outlined in **Response**
**2.14** below we have therefore included the results of the functional analyses based on TCGA
subtypes (**Fig. S6; Fig. 6b**). This provides further support for that the ATAC50-based
grouping more accurately predicts functional responses such as self-renewal, invasion, drug
sensitivity and tumorigenicity of the patient-derived GSCs.

**iv)** Both reviewers proposed to include more clinical data in our study, which we agree would
be beneficial but in practice is not an easy task. To address this we used the most obvious
and straightforward approach which was to include survival data for all glioblastoma patients.
They were stratified based on the ATAC60 clusters, which had showed a very high overlap
with ATAC50, and by adding ten more patients the ATAC stratification produced a strong
significant survival difference between C2 and C3 patients (**Fig. 6f**).

**Point-by-point responses to reviewers' comments**

**Reviewer #2, expert in ATAC-seq (Remarks to the Author):**

We thank R2 for thought-provoking and constructive comments. We have tried our best to
address and respond to them and have revised the manuscript accordingly.

In the manuscript by Lu et al. the authors performed a cross-species epigenome analysis of
mouse and human GSC cultures. The authors analyzed and compared the chromatin-
accessibility landscape of nine mouse GSC cultures of defined cell of origin and used sixty
patient-derived GSC cultures by assay for transposase-accessible chromatin using
sequencing (ATAC-seq). The authors have claimed to have uncovered a variability of both
mouse and human GSC cultures that was different from transcriptome analysis and better at
predicting functional subgroups.

Major Criticism:

1) Starting with the ATAC method, they used formaldehyde to fix it and then reverse
crosslink it after tagmentation which is something the communicating author used in a
previous ATAC-seq paper. For ATAC-seq I believe the need to fix the cell due to fluorescent
staining and visualization afterward, but here they are only using those tissue for bulk ATAC,
therefore, what's the precise reasoning for doing the formaldehyde crosslinking. Even if the
authors claim using their method wouldn't change the ATAC profile, most in the field still
prefer the option of capturing the chromatin in its native state without any crosslinking. Then
why is there any need to use this method? It seems to create problems and raises concerns.

**2.1.** We can understand this concern but we have previously shown that mild formaldehyde
treatment does not change the chromatin profiles of cultured cells (*Chen X et al, Nat Meth,*
*2016, PMID: 27749837*). The reason for using crosslinking in our study was to reduce
technical bias between the samples. In this way we could harvest cells at different time
points, since it was not possible to maintain and prepare 60 cultures at the same time. In
addition, two independent studies have also showed that mild formaldehyde fixation does not
interfere with tagmentation (*Payne A et al, Science, 2021, PMID: 22284301; Cusanovich D et*
*al, Nature, PMID: 29539636*). Taken together, we believe it is reasonable to perform ATAC-
seq with mild formaldehyde fixation.

2) This reader doesn't really understand the need for including the mouse data as aspect of
the study for Figure 1 and should remove this to strengthen the authors' rationales.

**2.2.** This is an important point which made us aware of the need to better describe the
rationale of our study. It builds on our previous cross-species transcriptome investigation
where we showed that a mouse cell-of-origin (MCO) gene signature could stratify patient-
derived glioblastoma cell cultures into three clusters, of which two were further investigated
and found to be functionally distinct (*Jiang Y et al, Cell Rep, 2017, PMID: 28122246*). The
MCO gene signature was derived from mouse GSC (mGSC) cultures of different origin, and
were included in this study since we wanted to analyze the relation of developmental origin
and epigenetic regulation in GSCs. Cross-species analyses of both transcriptomes and
epigenomes have been performed for many cancer types to obtain a better understanding of
how developmental biology contribute to cancer mechanisms (*Johnson RA et al, Nature,*
*2010, PMID: 20639864; Gibson P et al, Nature, 2010; PMID: 21150899; LaFave LM, Cancer*
*Cell, 2020, PMID: 32707078*), but this is, to our knowledge, the first cross-species analysis of
glioblastoma ATAC-seq data. In our revised manuscript, we have tried to better explain the
rationale of including the mGSCs in Introduction and Results.

This data appears to detract from their observations in the first figure of the manuscript. The
authors used cells from three different origins (used in triplicate representing 9 cell lines) and

performed bulk ATAC. One would predict that cells from different origins should definitely
cluster well together, and what was presented in Supplementary Fig. 1C basically represents
a problems using the triplicate or poor-quality reads.

**2.3.** This is a valid and understandable comment. Based on our previous transcriptome
analysis (*Jiang Y et al, Cell Rep, 2017, PMID: 28122246*) we had expected that mGSC
cultures of the same cell of origin would have clustered together. We were surprised by the
result but decided, based on the QC parameters, FRiP (fraction of reads in peaks), TSS
enrichment, Pearson correlations etc to move forward with the data. This comment, however,
by R2 instigated a closer look at the data. We then realized that there was a relation between
the clusters and the FRiP values (**Response fig. 1A**). To confirm or reject our previous
result, we performed new ATAC-seq analysis of all nine mGSC cultures with the Omni-ATAC
method (*Corces R et al, Nat Methods, 2017, PMID: 28846090*). The data turned out to be of
clearly higher quality based on multiple analyses (**Fig. 1b, c; Fig. S1a, b; Response fig.**
**1B**). PCA analysis now showed clusters of the same cell of origin (**Fig. 1d; Response fig.**
**1B**) and importantly, these were not connected to FRiP values (**Response fig. 1B**). Thus, we
decided to replace the mGSC ATAC-seq data with the newly generated Omni-ATAC-seq
data and revise the manuscript according to the new results (**Fig. 1; Fig. S1; Fig. 7; Fig. S7**)
including all text connected with those figures.

**Response figure 1.** FRiP and PCA of **A**) previous mGSC ATAC-seq data where we used a
cut-off of 10% (dashed blue line) for FRiP, and **B**) new mGSC Omni-ATAC-seq data where
we used a cut-off of 20% (red line) for FRiP.

While it is totally understandable since it's from different mice and only 3 in each group can
result in huge variation observed. The authors should either remove the outliers and increase
the number in each group to see if the PCA clustering improves. Figure as Supp1B basically
shows no significant clustering between those SCXX samples by Pearson correlation, but
they switch to NMF method to make it look more significant in Figure 1d.

2.4. The new higher quality mouse GSC Omni-ATAC-seq data produce robust cell of origin clusters.

In Figure 1C, the authors argue the peaks are specific to different origins, but in fact they are also trying to split replicates of each origin and mix them in groupings based on the NMF clustering.

2.5. Previous Fig. 1c has been revised based on the new Omni-ATAC-seq data and is now **Fig. 1f**. In that figure we show genome tracks of three genes (*Cdk5r1*, *Runx1*, *Kif5c*) each connected with unique ATAC peaks of the three mGSC groups (**Fig. 1e**).

Through Figures 1E and 1F is basically regarded as hacking the statistic analysis to fit their proposed model. A knowledgeable reader would say they could see a clear trend in Figure 1F between those different origins, even though they are not significant in p values, as well as shown in Figure 1E, the authors just switched some of the samples to fit it as statistically significant even though these two figure resemble each other. This is a problem in the readers view.

2.6. These figures have been removed from the revised manuscript since they do not represent the new mouse Omni-ATAC-seq data. We need, however, to point out that we did not move the previous data around randomly to obtain statistical significance. The previously presented two main clusters (A+B and C) were the result of three different clustering methods.

In Figure 1G the authors only displayed their new grouping to show some survival differences This then questions how does one explain about the difference survival analysis of the origins then? That too is also likely different and maybe more significant than the authors original intent.

2.7. These figures have been removed from the revised manuscript because they do not represent the new data. Survival analyses of cell-of-origin groups, both with regard to primary (injecting RCAS virus) and secondary (injecting mGSCs) orthotopic tumor development, have been investigated (*Jiang Y et al, Cell Rep, 2017, PMID: 28122246*), and were for both significantly different comparing mGC1_{GFAP} VS mGC2_{NES} or mGC3_{CNP}, but not between mGC2_{NES} and mGC3_{CNP}.

Found in Figure 1G are TFs the authors selected from Figure 1i, which is presumed to yield a difference in binding judging from the heatmap, but their plot showed quite the opposite. Most knowledgeable readers would not regard these binding differently as this is acknowledged merely by the insignificant p values.

2.8. Here we believe that R2 is referring to previous FigS1G which showed footprint analysis of TF binding. Correctly noted, that figure showed both significant (SLUG, SIX2, SIX4, TWIST, $p < 0.05$) and non-significant (NEUROD1, BRN2) footprints but unfortunately the legend for this panel was inaccurate which we apologize for.

In the revised manuscript we have omitted all results from footprint analysis using mouse data since we have realized that this analysis requires a sequencing depth of at least 200 million reads (*Schep A et al, Genome Res, 2015, PMID: 26314830*) which we do not obtain with our Omni-ATAC data.

3) For addressing the human analysis, the number of samples used and data itself are truly valuable, and capturing brain tissue is not an easy task, so there is strong appreciation for what was needed for them to generate those datasets, the QC looks OK, but this too still

widely varies between samples used. This is understandable since patient variability and their
collecting at different times by different personnel may represent the inconsistencies in some
of the QC standards. The variation in TSS enrichment and Frip scores would lead to the wide
variation in percent of promoter and enhancer peaks, where this is likely due to their
technical inconsistencies with this difficult tissue type, but this aspect should not be listed as
a biological difference and the focus in Figure 3.

**2.9.** We think that R2 refers to previous and current **Fig. 2** here where we display different
types of analyses of the ATAC-seq data across 60 patient-derived GSC cultures. Our
interpretation of the above comment is that R2 thinks that the heterogeneity displayed across
GSC cultures is due to technical bias and not a reflection of biology. To address that we have
divided our response in three parts:

**i) Samples**

We have used patient-derived glioblastoma stem cell cultures and not glioblastoma tissue.
This information about the samples was unfortunately absent in the Methods section of the
first submission and we apologize for that. The information has been included in the revised
manuscript under Methods.

We agree that tissue would have been much more difficult to analyze and probably would
have produced a larger variability, even between samples of the same patient because of
intra-tumor heterogeneity. Furthermore, the inter-patient heterogeneity would have been
affected not only by tumor cells but also by stromal cells, and also by tissue handling,
storage and preparation at different time points.

GSC cultures, however, lack stromal cells and were maintained and harvested under the
same conditions by a few people with extensive experience of maintaining and handling
these cultures. This has in our mind minimized the technical bias between samples.

**ii) ATAC-seq data quality**

In previous and current **Fig. S2**, which has not been changed in the revised manuscript, QC
analyses including FRiP, TSS enrichment and Pearson correlation of technical replicates are
shown and we believe, in accordance with R2's comment above ("QC looks OK"), that the
human data quality is fine.

**iii) Inter-culture heterogeneity across 60 patient-derived cultures**

Our study presents, to our knowledge, the largest patient-derived GSC cohort that has been
analyzed with ATAC-seq. We think that **Fig. 2** contributes to illustrate the inter-patient
heterogeneity of our cultures, which is important since glioblastoma is well-known for being a
highly inter-patient heterogeneous disease. It provides strong support for that our human
data is relevant and representative to glioblastoma.

(Figure 3, panels f,g,h doesn't provide much real information). This questions the value in
using this information.

**2.10.** We agree with R2 that some of these figures could be better suited in the
supplementary information and have moved previous Fig. 3f to **Fig. S3f**. The purpose of this
figure is to illustrate the impact of promoter versus distal regulatory element regions to define
the ATAC50 clusters.

Figure 4, panels a,c,d is still question those distribution differences by ATAC peaks
shown, even though called "significant" there is not much difference by eye, and this too
could be due to technical variation.

**2.11.** We agree that these statistical differences are not immediately obvious to the eye. One
reason is that the analyses are based on large data sets. We have re-analyzed the statistical
calculations and can confirm that the previous analyses were correct. To clarify, **Fig. 4a** is
not based on ATAC-seq data but on gene expression array data for the human GSC
cultures.

The authors used the same NFM clustering 50 human samples which doesn't correlate well
with TCGA subtypes and mildly correlate with MCO scoring. What the authors claimed is
their ATAC is yields a better predictive value and can capture sample difference that can't be
found in the RNA-seq results, this is not believable based on the data presented thus far.
From what we know, RNA-seq has less technical variation and should be more robust to
present sample differences.

**2.12.** Based on our arguments in **response 2.9** above we do not believe that our cluster
result is a consequence of technical variation. Glioblastoma has been molecularly stratified
by many methods of which transcriptome and methylome have been most frequently used
(*Brennan et al, Cancer Cell, 2013, PMID: 24120142; Ceccarelli et al, Cell, 2016, PMID:*
*26824661*). They have shown some but far from complete overlap. In a very recent study an
integrated molecular analysis of 99 glioblastoma tumors was presented based on ten
different analyses, including genomics, transcriptomics, methylomics, proteomics, lipidomics
and metabolomics (*Wang et al, Cancer Cell, 2021, PMID: 33577785*). This displayed some
overlap but also very different clusters between the different data sets.

The fact that our ATAC-seq data does not reproduce the clusters of TCGA subtypes goes
well in line with the above and with a recently published ATAC-seq paper (*Guilhamon P et al,*
*eLife, 2021, PMID: 33427645*) that was also cited in the original manuscript.

Whatever marks survival, whether through sphere growth or differences in drug response,
this reader believes you can find a similar trend using any other random grouping method
that wouldn't carry any significant biological meaning. The way the authors are trying data
mining with the data shown that doesn't not cluster well with real subtypes is problematic. If
their 50 patients NMF clusters matches poorly with TCGA data, then this too is a problem.
The authors are effectively claiming their datasets are better predictor than all the TCGA data
combined. If by comparison the authors would need much more experimental and clinical
data to back up their claims for the groupings shown.

**2.14.** Continuing where we left in **response 2.13**, we do not find it problematic or surprising
that the ATAC clusters do not agree with the TCGA subtypes. In our view the glioblastoma
field has for a number of years been moving away from the TCGA subtypes. One reason
being that they have not proven clinically informative and were for example not integrated in
the revised 2016 WHO classification of glioblastoma, as opposed to for example the
molecular classification of medulloblastoma (*Louis DN et al, Acta Neuropathol, 2016, PMID:*
*27157931*). Another being that the expanded knowledge from single cell RNA-seq has
shown that glioblastoma cells are highly diverse and dynamic and occur in different states
believed to be developmentally dictated (*Neftel C et al, Cell, 2019, PMID: 31327527;*
*Richards LM et al, Nat Can, 2021*).

Furthermore, we need to stress that we have not performed data mining. We have simply
used the ATAC50 groups obtained from the NMF cluster analysis when analyzing the
functional data in previous and current **Fig. 5**. However, this comment from R2 highlighted
the usefulness to present the same data grouped by TCGA subtypes as a reference to
ATAC50 (**Fig. 5; Fig. 6a**). These analyses have been included in the revised manuscript as
**Fig. S6** and **Fig. 6b**. When comparing the two classifications we find that ATAC50 is superior
to TCGA at predicting self-renewal, invasion, drug response and tumorigenicity compared
with TCGA subtypes.

Also, as pointed out by R2, the study would gain from more clinical data so we decided to
include all patients and use the ATAC60 clusters (which are highly similar when comparing to
ATAC50) in the survival analysis. By doing so we captured a significant survival difference
between ATAC60 C2 and C3 patients (**Fig. 6f**) which is particularly interesting since these
two clusters are the most similar, both in terms of molecular and functional data.

A reader would have argued that the authors should have focused more on the MCO score
correlation and strengthen a story around that concept, where at the very minimum this
would at least show readers that ATAC-seq data can be used to evaluate/correlate with MCO
and find strong biological relationships more effectively, at least some of the TF activity in
Figure 4h does look difference, or do a integrate RNA and ATAC analysis using MCO
grouped RNA-data with their ATAC data and see how the different MCO group glioblastoma
regulome works, that would be more interesting than those promoter and enhancer
percentage QC and correlation analysis.

**2.15.** We think that this comment is relevant and interesting. We had previously showed with
transcriptome analysis that the mGSC cultures of different origin clustered with either
proneural (mGC1_{GFAP}, mGC3_{CNP}) or mesenchymal (mGC2_{NES}) TCGA classified glioblastomas
(*Jiang Y et al, Cell Rep, 2017, PMID: 28122246*). With the new mouse ATAC-data we could
produce the same cell of origin groups (**Fig. 1d**). Analyses of annotated genes (**Fig. 1e**) and
enriched TF motifs of unique ATAC peaks for each mGSC group (**Fig 1g; Fig. S1g-i**)
sustained this relationship. We then performed a cross-species analysis of significantly
enriched TF motifs of each mGSC group (**Fig. S1g-i**) with all significant TF motifs of ATAC50
cluster-unique ATAC peaks (**Fig. S5b**). We found a positive correlation for a number of TFs
(**Fig. S7a**) and this corroborated the alignment of mGSC groups along a proneural to
mesenchymal axis and a relationship between mGC1_{GFAP} and ATAC50 C1, mGC3_{CNP} and
ATAC50 C3, and mGC2_{NES} and ATAC50 C2, strongly proposing an important role of
neurodevelopmental regulation of glioblastoma.

In addition, we showed already in the previous version of the manuscript (**Fig. 7**) a cross-
species analysis of mouse and human ATAC data where we compared MCO and ATAC50
clusters. This figure remains in the revised version and has been updated with the new
mouse ATAC-seq data in **Fig. 7d-e**. The take-home message is that there is a considerable
overlap of the MCO and ATAC50 classifications (**Fig. 7a**), which is not dictated by analyzing
ATAC peaks of promoter regions (**Fig. 7b**) or distal regulatory element regions (**Fig. 7c**) of
MCO genes. However, using the unique mouse ATAC peaks of the new Omni-ATAC data
(**Fig. 1e**) and converting these, through two steps of peaks-to-genes analyses, to human
ATAC peaks, could with higher precision than MCO classification predict the ATAC50
clusters (**Fig. 7d**). Analyzing the enriched TF motifs of the mouse (**Fig. 7e**) and human (**Fig.**
**7f**) ATAC peaks generated in **Fig. 7d** showed a 90% overlap in the top-10 TFs, which
support the existence of a species-conserved developmental regulation of glioblastoma.

Overall evaluation is that the authors would at the very least need a major revision and to
rethink about what and how they want to analyze this truly valuable dataset. What they are
claiming now remains arbitrary and is weak without further biological and clinical data to back
up their findings and observations.

**2.16.** We have very carefully considered all comments from R2 and have used them as a
basis to revise our manuscript. With our unique mouse and human data sets and cross-
species approach we present a novel functional stratification that can also predict patient
survival. The cross-species analysis support that the clusters are driven by species-
conserved, neurodevelopmental mechanisms. We hope that our findings will provide
stepping stones for further investigations to find targetable mechanisms in these or even
more refined subsets of glioblastoma.

**Reviewer #3, expert in glioblastoma stem cells (Remarks to the Author):**

We thank R3 for relevant and constructive comments. We have tried our best to address and
respond to them and have revised the manuscript accordingly.

In the manuscript 'Conserved cell-lineage controlled epigenetic regulation in human and
mouse glioblastoma stem cells determines functionally distinct subgroups', Lu et al. describe
a novel approach for classifying GSC cultures based on transposase-accessible chromatin
distribution. They identified three independent ATAC-seq clusters in 9 murine and 60 human
GSC cultures that were different compared to gene expression-driven approaches. In their
mouse model, they found functional differences in sphere-forming capacity, in vivo
aggressiveness, drug response, and transcription factor motifs in the ATACs. They further
went on performing similar experiments with human GSC cultures. The sequences were
predominantly DREs in both species. Also, in the human system, they report functional
differences between the clusters in terms of GSC clonogenicity and drug response. The
human ATAC-seq clusters overlapped with the MCO stratification better when compared to
the murine GSCs.
Importantly, individuals belonging to two clusters showed significantly different survival in a
small cohort of patients. Finally, they analyzed the TF motifs in the murine and human GSC
ATAC-seq samples and found a significant overlap of these motifs.

The manuscript is well written and describes an in-depth analysis of ATAC-seq data obtained
from a big collection of GSC cultures and shows that the patients can be clearly divided into
well-defined clusters with this method. However, it is a retrospective study with few
biologically functional data. Thus, the practical value of this approach in the classification of
GB over the current gene expression-driven methods is not entirely evident.

**3.1.** We agree with R3 that our findings will not result in any changes in clinical practice of
glioblastoma patients. Our study is an attempt to understand the underpinnings of
developmental regulation of glioblastoma and our findings thus far support that chromatin-
accessibility is superior to gene expression in predicting functional responses of glioblastoma
cells and that these responses seem to be neurodevelopmentally regulated and controlled by
distal rather than promoter regions. Continued studies will be necessary to understand the
underlying mechanisms, which we believe are outside the scope of this investigation.

However, we would like to argue that the practical (clinical) value of TCGA subtypes or any
other molecularly-based glioblastoma classification have not proven clinically useful as yet
although some have been around for a decade or more. This proposes that other molecular
approaches are needed, such as for example ATAC-seq.

In addition, as a consequence when responding to comment 4) below, we have in the
revised manuscript included a survival analysis of all glioblastoma patients using the
ATAC60 classification, which interestingly produced a significant difference between patients
of ATAC60 C2 and C3 (**Fig. 6f**).

Major points:

1) In the mouse model (figure 1), the authors use ATAC-seq to divide the GSC cultures into
three distinct and clearly separated clusters. Then they use this scheme to perform their first
functional tests (e-h) to later unite clusters A and B in Figures i and j. This seems a bit forced.
If the authors suggest ATAC-seq as the basis for grouping the GSC cultures and three clear
ATAC-seq clusters were identified, the differences within these three clusters should be
analyzed. Given the investigation's retrospective nature, this grouping should not be changed

depending on the experimental question to be answered.

**3.2.** This is a valid and understandable comment which was raised also by R2. Because of
this we scrutinized the previous mouse ATAC data and realized that there were problems
with quality and variability among the mouse GSC (mGSC) samples. Please, see **Response**
**2.3** and **Response figure 1** above for more detailed information, but in summary, we
performed new ATAC-seq analysis of the nine mGSC cultures which produced data of higher
quality and a PCA separation into cell of origin groups, in line with the previous
transcriptome-based PCA (*Jiang Y et al, Cell Rep, 2017, PMID: 28122246*). Because of the
significantly higher quality of the new data we decided to use that instead and as a
consequence all results that included mouse ATAC data were revised (**Fig. 1; Fig. S1; Fig.**
**7; Fig. S7**) and connected texts edited in the revised manuscript.

Regarding the previous mouse ATAC data we would still like to stress, to remove any
suspicions about data mining, that what we showed was the result of three different cluster
methods: NMF (**previous Fig. 1d**) and hierarchical clustering (**previous Fig. S1d**) identified
the same three clusters (A: SC64, SC81, SC83; B: SC84, SC112; C: SC37, SC50, SC52,
SC74), while PCA (**previous Fig. S1c**) produced a slightly different result where one culture
(SC83) moved from cluster A to cluster B.

2) In the sphere-forming assay (figure 1e and f), the behavior of the murine GSC cultures
grouped by MCO and ATAC-seq are compared. The analyses show only differences for the
ATAC-seq clusters. The authors describe that the system of murine GSCs was published
before (*Cell Reports* 18, 977–990, January 24, 2017). How can it be explained that a similar
analysis done in the cited paper showed highly significant differences between mGC1(GFAP)
and mGC2(NES) or mGC3(CNP), while the new analyses did not?

**3.3.** This is a highly relevant question. Before answering it we would like to point out that with
the new mouse ATAC data that produced cell of origin groups we did not think that this
analysis was important and have removed it from the revised manuscript.

To the explanation: In *Jiang Y et al, Cell Rep, 2017* the primary spheres were derived from
freshly dissociated mouse glioblastoma tissues (*Jiang Y et al, Figure 3A-C*), and secondary
spheres were derived from the primary spheres (*Jiang Y et al, Figure 3D*). Thus, these
analyses were done on acute mouse glioblastoma samples. The analysis showed that
mGC_{2NES} and mGC_{3CNP} cells could not be maintained as spheres, so all subsequent
analyses of these cells were performed on adherent cultures. In the previous manuscript (*Lu*
*X et al*) the sphere assay (**previous Fig. 1e, f**) was performed on adherent cell cultures
below passage 13. This result would be more comparable to the ELDA result (*Jiang Y et al,*
*Figure 3E*), where there was a low significant difference between mGC1_{GFAP} and mGC2_{NES} or
mGC3_{CNP}. Although not significant in previous *Lu X et al*, there was still the same trend.

3) Critics concerning data selection apply to the data obtained for the human GSC samples.
In figure 2, the authors describe the identification of robust clusters with 60 GSC samples
(ATAC60). It is not entirely clear why they remove 10 samples for further analyses. In the
text, this procedure is explained by the lack of genetic data for the 10 lacking samples but
with such a strong bioinformatic background these analyses should be possible and done.

**3.4.** We completely understand this critique. We did consider to perform RNA-seq analysis
on all 60 samples so that we would have complete gene expression and ATAC-seq data for
all 60 cultures, but the honest reality is that we have limited grants and needed to prioritize
which analyses to perform. The strongest argument for settling with 50 human GSC (hGSC)
samples in most of the analyses (those where we wished to relate the data to MCO and
TCGA) was because the clustering was so robust between ATAC50 and ATAC60, where
only 3 of 50 cultures changed cluster (U3013MG: ATAC60-C2, ATAC50-C1; U3060MG:

ATAC60-C3, ATAC50-C2; U3198MG: ATAC60-C3, ATAC50-C1). This made us decide to
use the available mGSC array data and prioritize other analyses.

4) While the xenograft data in figure 6 is convincingly showing that mice injected with human
GSCs belonging to different ATAC-seq clusters have different survival rates, the patient data
is not so strong. This may be at least partially due to the low number of patients. It may be
helpful to include the patients for the ATAC60 panel to strengthen the significance (this could
even be done without further information concerning the gene expression profiling).
Especially because the different survival between two clusters is also underlined in the
abstract.

**3.5.** This is an excellent suggestion! We performed survival analysis of all patients and used
the ATAC60 clusters which showed a significant difference between C2 and C3 patients
(**Fig. 6f**). This is especially interesting since these two clusters are the most similar, both
molecularly and functionally.

5) The reviewer has not fully understood the discrepancy of the temozolomide sensitivity
measured in Figures 6f and 5 e-g. If these differences are too subtle to be detected in AUC
analyses, it may be enough to explain the discrepancies for the survival in the mouse model
and the patients. The authors may test their hypothesis in vivo for some of their cultures to
better mimic the patients' situation.

**3.6.** In previous and current **Fig. 5e-g** we show pairwise comparisons of areas under the
dose-response curves (AUC) for each drug tested. In **previous Fig. 6f** we showed the actual
dose-response curves for each cluster for temozolomide (that are the basis of the analysis in
Fig. 5e-g). Although TMZ AUC values for the three clusters were not significantly different,
viability at individual doses in the dose-response curve can still be.

In the previous manuscript we showed Fig. 6f in support of our hypothesis that the improved
survival of C3 patients could be due to a better response to treatment. However, we have
decided to omit this figure in the revised manuscript because the concentrations that
produced the significant differences (62-500 μ M) in the cells were vastly higher than the
predicted concentration of 15-35 μ M in a TMZ treated patient tumor (*Strobel et al,*
*Biomedicines, 2019*).

6) The finding that murine and human GSC ATAC clusters are largely overlapping is very
interesting but may be biased by the growth conditions. The GSC cultures are established
and kept in defined media with dominant growth factors (FGF2 and EGF). These could lead
to the activation of selected signaling pathways in both murine and human GSCs that, in
turn, may result in the activation of similar transcription factor patterns. This should be
investigated carefully because TFs of the AP1 family are downstream substrates of MAP
kinases that are known to be strongly activated by growth factor signaling. Thus, the authors
may control if the culture conditions influence the ATAC-seq data.

**3.7.** This is a highly appropriate reasoning and when addressing it we realized that
information about both mouse and hGSC cultures was absent from Methods. We do
apologize for this and have added this crucial information in the revised version.

To respond to the comment, it is correct that the patient-derived GSCs have been cultured in
presence of exogenous EGF and FGF2. However, our mGSC cultures have from
explantation and throughout the project always been cultured in absence of any added
growth factors. We found, a decade ago, that if EGF and FGF2 were added to the media the
mGSC cultures were overgrown by normal NSCs (*Jiang Y et al, Neoplasia, 2011, PMID:*
*21677873*).

Minor points:

1) From the M+M part, it is not clear how the sphere formation assay was done. Were the primary spheres established from in vitro cultures or from freshly dissociated xenografts? If they were established from in vitro cultures (which could explain the identical rates of primary, secondary, and tertiary spheres), it is a bit misleading to describe them as primary spheres. In this case, it would be better to rate it as in vitro proliferation.

3.8. We are sorry for the lack of information. All sphere assays were performed on established cultures below passage 13 for mouse and below passage 20 for human cells.

2) The same question applies to the ELDA and the proliferation analyses. Were the cells from freshly dissociated tumors or from GSC in vitro cultures?

3.9. Again, sorry for the lack of information. ELDA and proliferation have been performed on established cultures below passage 20. We have clarified this in Methods and added passage numbers.

3) For the survival analyses in figure 6a, the mice were inoculated with different numbers of cells. Although this becomes evident from the supplementary table, it should be clearly stated in the text.

3.10. Excellent point. We have clarified that in legends for **Fig. 6a**.

Reviewers' Comments:

Reviewer #2:

Remarks to the Author:

The manuscript by Lu et al. entitled, "cell-lineage controlled epigenetic regulation in glioblastoma stem cells determines functionally distinct subgroups and predicts patient survival" appears to have been adequately revised to an acceptable degree. However, some concerns and issues remain as noted below.

Even though this reviewer remains a little bothered by the formaldehyde crosslinking steps used, the authors do reason that they checked all the QC to make sure the read quality was assured. As well, the authors did re-generate and repeat the mouse ATAC Seq studies again and indicates that the quality is better this time.

The authors were responsive and removed the figures confusing to readers from the previous version and have added new better quality figures thereby this further clarifies some of the ambiguous data described previously.

While the significance of human data still needs more exploration to make more profound conclusions, this reviewer understands the limitations in acquiring and substantiating the human data. The authors do provide better mouse data and therefore understood why the authors emphasized the mouse xenograft data and how human and mice -ATAC Seq might become more highly correlated to better bridge mouse and human models and some of the mouse functional data to strengthen the analysis.

Minor questions-

In Fig. 4 the authors display the box plots (panels a, c, and d) based on the heatmap in panel b for C1, 2, &3 against GBM tumor types. However, it appears the degree of change (Δ s) are not so dramatic, despite the statistical significance noted. A more nuance explanation would help the reader appreciate why this maybe the case, especially for panels c and d.

Is it possibly due to patient variance, data quality, source of samples and sampling preparations, etc.? Moreover, this questions some of the correlations presented in panel g, or lack thereof, made from individual TF footprints noted in panels h. It appears that only JUNB and FOSL2 are the only ones relevant and this should be pointed out clearly to the reader. Nonetheless, these are minor points.

Overall, the changes have certainly strengthen the manuscript and believe it provides stronger impact from the previous version.

Reviewer #4:

Remarks to the Author:

This is an interesting study, which performs a cross-species analysis from a large number of patient-derived glioblastoma stem cell (GSC) cultures and mouse GSC cultures with genetically defined cell-of-origin, and discovers a small but significant developmental cell-of-origin ATAC-seq signature in human GBM, with potential prognostic significance. The revised manuscript addresses many of the reviewers' comments and shows improved technical replicates in mouse ATAC-seq GSC data related to cell-of-origin, which bolsters the premise of the study and its conclusions. Furthermore, the re-analysis now demonstrates a significant survival signature within ATACseq60 C2 and C3 clusters, using retroactive patient survival data, which corroborates the authors' prior mouse xenograft data, and serves as an important finding to be explored in future studies for molecularly-based classification of GBM using

ATAC-seq data. Many of reviewer 3's concerns have been satisfactorily addressed with this new analysis. However, some of them were not addressed fully and continue to pose concern. As requested, below I provide my itemized assessment of the authors' response to reviewer 3's critiques.

3.1 (addressed)

The authors now show a survival analysis using human GBM patient survival data, from whom hGSCs were derived, using ATAC60 classification (~to ATAC50 but containing 10 more patients), which allowed them to detect significant difference in patient survival within the C2 and C3 cluster signature. This is a nice complement to the already existent orthotopic xenograft data in Figure 6, which shows significant survival difference stratification based on ATAC50 and ATAC60 C2 vs. C3 cluster signatures.

3.2 (addressed, needs clarification)

The new mouse ATAC-seq data analysis using Omni-ATAC-seq shows robust technical replicate clustering on PCA and acceptable FriPs. One important note – the authors did not clearly mention whether they also re-analyzed their human hGSC ATAC-seq data using Omni-ATAC-seq; this should be clarified in the text beyond simply adding the reference #34. As the data is being compared, both human and mouse ATAC-seq analyses should be performed using the same pipeline.

3.3 (remaining concern)

The explanation as to the differences between the published analysis and the previous data shown is logical, but it raises concerns about the effect of passage number and culture conditions, and whether the uncovered biomarkers are broadly clinically relevant. At the very least, the authors should include passage number and culture conditions for all their hGSC cell lines in Data Table 1, and use these as covariates in additional survival analyses.

3.4 (remaining concern)

Given that the authors end up finding survival benefit with ATAC60, but not with ATAC50, Reviewer3's critique about data selection is even more relevant. I agree with reviewer 3 that the lacking genetic analyses should be done and ATAC60 should be explored in parallel to ATAC50. Or, since ATAC50 and ATAC60 are so similar, they can show only analyses performed using the clinically-relevant ATAC60 signature.

3.5 (addressed)

This has been satisfactorily addressed, see 3.1

3.6 (remaining concern)

The authors did not attempt to test their hypothesis regarding TMZ sensitivity *in vivo*, as reviewer 3 had suggested, and their explanation for now omitting Figure 6f is not sufficiently well justified.

3.7 (addressed)

The authors have clarified that mouse GSCs are not cultured using EGF and FGF2. While it is still possible that some of their findings are biased by growth conditions, since this study is largely cell culture-based, further analysis of TFs feels beyond the scope of their study.

3.8 (addressed, needs clarification)

The authors have clarified the passage numbers for human GSCs as less than 20 passages, which some researchers consider to still recapitulate primary GBM biology and deem as patient-derived "primary" cell culture (especially if maintained within serum-free NB conditions, PMID: 16697959) while others suggest may already be considered "established" (PMID: 30894629). Since both the mouse and human GSCs were grown at passage greater than 10, comparison is justified. Of note, the study would have been more impactful in regards to true cell of origin, if both mouse and hGSC were compared at early passage (less than 5), and, even more so, in acutely dissociated conditions. As suggested in other comments, including the actual passage used for each line, and analysis of correlation between cell or origin score to extent of passaging, would be insightful, and could further address some of reviewer 3's concerns.

3.9 (addressed)

Satisfactory clarifications have been made.

3.10 (addressed)

Satisfactory clarifications have been made.

REVIEWER COMMENTS

We thank both reviewers for their constructive comments to further improve our manuscript. We were glad to see that most were clarified in the previous revision and have addressed the remaining concerns below, point by point.

Reviewer #2 (Remarks to the Author):

The manuscript by Lu et al. entitled, "cell-lineage controlled epigenetic regulation in glioblastoma stem cells determines functionally distinct subgroups and predicts patient survival" appears to have been adequately revised to an acceptable degree. However, some concerns and issues remain as noted below.

Even though this reviewer remains a little bothered by the formaldehyde crosslinking steps used, the authors do reason that they checked all the QC to make sure the read quality was assured. As well, the authors did re-generate and repeat the mouse ATAC Seq studies again and indicates that the quality is better this time.

The authors were responsive and removed the figures confusing to readers from the previous version and have added new better quality figures thereby this further clarifies some of the ambiguous data described previously.

While the significance of human data still needs more exploration to make more profound conclusions, this reviewer understands the limitations in acquiring and substantiating the human data. The authors do provide better mouse data and therefore understood why the authors emphasized the mouse xenograft data and how human and mice -ATAC Seq might become more highly correlated to better bridge mouse and human models and some of the mouse functional data to strengthen the analysis

Minor questions-

In Fig. 4 the authors display the box plots (panels a, c, and d) based on the heatmap in panel b for C1, 2, &3 against GBM tumor types. However, it appears the degree of change (Δ s) are not so dramatic, despite the statistical significance noted. **A more nuance explanation would help the reader appreciate why this maybe the case, especially for panels c and d. Is it possibly due to patient variance, data quality, source of samples and sampling preparations, etc.?**

Response R2 - 1. The point is well taken. In the revised manuscript, we have added a sentence in conjunction with **Fig. 4a, c, d** explaining the overlapping box plots, lines 250-252: "*Although the differences between ATAC50 groups in most of the comparisons in Fig. 4a, c, d were statistically significant the box plots were still highly overlapping reflecting the extensive intertumor heterogeneity of GBM.*"

Moreover, this questions some of the correlations presented in panel g, or lack thereof, made from individual TF footprints noted in panels h. It appears that only JUNB and FOSL2 are the only ones relevant and this should be pointed out clearly to the reader.

Response R2 - 2. This is a good point. All TF footprints in **Fig. 4h** are significant ($p < 0.05$) and exact p-values have now been added to each revised graph. In addition, we have also changed the order of the graphs in Fig. 4g-h to improve readability, so that the same TFs are on top of each other.

Nonetheless, these are minor points. Overall, the changes have certainly strengthen the manuscript and believe it provides stronger impact from the previous version.

Reviewer #4 (Remarks to the Author):

This is an interesting study, which performs a cross-species analysis from a large number of patient-derived glioblastoma stem cell (GSC) cultures and mouse GSC cultures with genetically defined cell-of-origin, and discovers a small but significant developmental cell-of-origin ATAC-seq signature in human GBM, with potential prognostic significance. The revised manuscript addresses many of the reviewers' comments and shows improved technical replicates in mouse ATAC-seq GSC data related to cell-of-origin, which bolsters the premise of the study and its conclusions. Furthermore, the re-analysis now demonstrates a significant survival signature within ATACseq60 C2 and C3 clusters, using retroactive patient survival data, which corroborates the authors' prior mouse xenograft data, and serves as an important finding to be explored in future studies for molecularly-based classification of GBM using ATAC-seq data. Many of reviewer 3's concerns have been satisfactorily addressed with this new analysis. However, some of them were not addressed fully and continue to pose concern. As requested, below I provide my itemized assessment of the authors' response to reviewer 3's critiques.

3.1 (addressed)

The authors now show a survival analysis using human GBM patient survival data, from whom hGSC were derived, using ATAC60 classification (~to ATAC50 but containing 10 more patients), which allowed them to detect significant difference in patient survival within the C2 and C3 cluster signature. This is a nice complement to the already existent orthotopic xenograft data in Figure 6, which shows significant survival difference stratification based on ATAC50 and ATAC60 C2 vs. C3 cluster signatures.

3.2 (addressed, needs clarification)

The new mouse ATAC-seq data analysis using Omni-ATAC-seq shows robust technical replicate clustering on PCA and acceptable FriPs. One important note – **the authors did not clearly mention whether they also re-analyzed their human hGSC ATAC-seq data using Omni-ATAC-seq; this should be clarified in the text beyond simply adding the reference #34.** As the data is being compared, **both human and mouse ATAC-seq analyses should be performed using the same pipeline.**

Response R4 - 3.2. Thank you for pointing this out. We realized that we had not described the ATAC-seq methods and data analyses clearly enough in Methods. We have now clarified this in lines 673-686 and 703. The essence of the additional information is that Omni-ATAC-seq was only used for mouse GSCs (since the human data already were of high enough quality), and the same bioinformatics pipeline was used for both mouse and human ATAC-seq data.

3.3 (remaining concern)

The explanation as to the differences between the published analysis and the previous data shown is logical, but it raises concerns about the effect of passage number and culture conditions, and whether the uncovered biomarkers are broadly clinically relevant. At the very least, the authors should **include passage number and culture conditions for all their hGSC cell lines in Data Table 1, and use these as covariates in additional survival analyses.**

Response R4 - 3.3. This is a good point. In our revised manuscript, passage numbers and culture conditions for all cultures included in the study have been added to **Data Table 1**. We were not completely clear of how to relate the passage numbers of the hGSC cultures in

Data Table 1 to survival. However, to further understand if passage number could have influenced our results in the functional analyses, we have compared passage numbers between the groups used in the functional analyses (**Fig. 5**, **Supplemental Fig. 6**, and new **Supplemental Fig. 7**), i.e. between ATAC50 groups (**Rebuttal Fig. 1a**), TCGA groups (**Rebuttal Fig. 1b**), and ATAC60 groups (**Rebuttal Fig. 1c**). We used one-way ANOVA comparing all three groups and t-test between pairs of groups, and could only find one significant difference in passage number between CL and MS groups in the drug response analysis (**Rebuttal Fig. 1b**). By that we feel confident that differences in passage number have not biased our results.

Rebuttal Figure 1. Comparisons of passage numbers of the cell cultures used in the different functional analyses (sphere formation, ELDA, proliferation, invasion, drug

response). One-way ANOVA and Student's t-test were performed on all graphs. (a) Groups based on ATAC50. All comparisons were non-significant. (b) Groups based on TCGA. CL vs. MS in drug response analysis was significant using t-test, *p=0.05. (c) Groups based on ATAC60. All comparisons were non-significant.

3.4 (remaining concern)

Given that the authors end up finding survival benefit with ATAC60, but not with ATAC50, Reviewer3's critique about data selection is even more relevant. I agree with reviewer 3 that the lacking genetic analyses should be done and ATAC60 should be explored in parallel to ATAC50. **Or, since ATAC50 and ATAC60 are so similar, they can show only analyses performed using the clinically-relevant ATAC60 signature.**

Response R4 - 3.4. We agree that we should have analyzed the functional data using ATAC60. Thus, we reanalyzed the data from **Fig. 5** and **Fig. 6a** using ATAC60 which is presented in the new **Supplementary Fig. 7** introduced after showing patient survival of ATAC60 groups, lines 363-370. The changes from ATAC50 to ATAC60 were small:

- **Fig S7c**, C1 vs C2 changed from ****p<0.0001 to ***p = 0.00015.
- **Fig S7d**, C1 vs C2 changed from *p=0.02 to **p=0.0034, and for C2 vs C3 from *p=0.014 to **p=0.0051.
- **Fig S7f**, number of significantly more sensitive drugs in C1 vs. C3 changed from 5 to 4
- **Fig S7g**, number of significantly more sensitive drugs in C3 vs C2 changed from 4 to 2
- **Fig S7h**: higher significant difference between C2 and C3, **p=0.0062 changed to ***p=0.00037

In summary, ATAC60 groups provided slightly higher significance for differences in invasion and in vivo tumorigenicity, while slightly lower for proliferation and drug response phenotype. We had added our new analysis in the main text (lines 363-370).

3.5 (addressed)

This has been satisfactory addressed, see 3.1

3.6 (remaining concern)

The authors did not attempt to test their hypothesis regarding TMZ sensitivity in vivo, as reviewer 3 had suggested, and their explanation for now omitting Figure 6f is not sufficiently well justified.

Response R4 - 3.6. To perform animal experiments must be ethically justified and to all extent possible preceded by supportive *in vitro* data. Here we did not think that the *in vitro* data was strong enough to initiate a treatment study since there were no significant differences in drug response at the clinically relevant concentrations which have been reported to be between 1-35 mM (Rosso L et al, *Cancer Res*, 2009 PMID: 19117994; Stepanenko AA et al, *Biomedicines*, 2019, PMID: 31783653). We do however think that one contributing factor to the survival differences between ATAC60 C2 and C3 patients may be differences in response to oncologic treatment (temozolomide and irradiation) and this is something we wish to follow up in future studies.

3.7 (addressed)

The authors have clarified that mouse GSCs are not cultured using EGF and FGF2. While it is still possible that some of their findings are biased by growth conditions, since this study is largely cell culture-based, further analysis of TFs feels beyond the scope of their study.

3.8 (addressed, needs clarification)

The authors have clarified the passage numbers for human GSCs as less than 20 passages, which some researchers consider to still recapitulate primary GBM biology and deem as patient-derived “primary” cell culture (especially if maintained within serum-free NB conditions, PMID: 16697959) while others suggest may already be considered “established” (PMID: 30894629). Since both the mouse and human GSCs were grown at passage greater than 10, comparison is justified. Of note, the study would have been more impactful in regards to true cell of origin, if both mouse and hGSC were compared at early passage (less than 5), and, even more so, in acutely dissociated conditions. As suggested in other comments, **including the actual passage used for each line, and analysis of correlation between cell or origin score to extent of passaging, would be insightful, and could further address some of reviewer 3’s concerns.**

Response R4 - 3.8. We agree with reviewer 4 that optimally we would have used similar and low passage cells for all our analyses. However, this poses a big problem when it comes to generating the numbers needed for technical and biological replicates, and was for our study with a large number of different cultures and extensive functional and phenotypic analyses not possible.

We have, per **Response R4 - 3.3**, added passage numbers for all cultures included in **Data Table 1** and also analyzed if the results could have been biased by differences in passage number but cannot find any evidence pointing to that (**Rebuttal Fig. 1**).

3.9 (addressed)

Satisfactory clarifications have been made.

3.10 (addressed)

Satisfactory clarifications have been made.

Reviewers' Comments:

Reviewer #4:

Remarks to the Author:

The authors have responded satisfactory to my remaining questions / concerns.

NCOMMS-21-05116B
Point-by-point response

Overview

We thank the Reviewers for their positive feedback on our manuscript.

Reviewer #4 (Remarks to the Author):

The authors have responded satisfactory to my remaining questions / concerns.

Thank you for your positive feedback.